# Dual blockade of CD47 and HER2 eliminates radioresistant breast cancer cells

Demet Candas-Green[1,14], Bowen Xie[1,2,14], Jie Huang [1], Ming Fan[1], Aijun Wang [3], Cheikh Menaa[1], Yanhong Zhang[4], Lu Zhang[1,2], Di Jing[1,2], Soheila Azghadi[1], Weibing Zhou[1,2], Lin Liu[1], Nian Jiang[1], Tao Li[1], Tianyi Gao[1], Colleen Sweeney[5,6], Rulong Shen[7], Tzu-yin Lin[5], Chong-xian Pan[6,8], Omer M. Ozpiskin[1], Gayle Woloschak[9], David J. Grdina[10], Andrew T. Vaughan[1,6], Ji Ming Wang[11], Shuli Xia[12], Arta M. Monjazeb[1,6], William J. Murphy [6,13], Lun-Quan Sun[2], Hong-Wu Chen[5,6], Kit S. Lam[5,6], Ralph R. Weichselbaum[10] & Jian Jian Li [1,6✉]

Although the efficacy of cancer radiotherapy (RT) can be enhanced by targeted immunotherapy, the immunosuppressive factors induced by radiation on tumor cells remain to be identified. Here, we report that CD47-mediated anti-phagocytosis is concurrently upregulated with HER2 in radioresistant breast cancer (BC) cells and RT-treated mouse syngeneic BC. Co-expression of both receptors is more frequently detected in recurrent BC patients with poor prognosis. CD47 is upregulated preferentially in HER2-expressing cells, and blocking CD47 or HER2 reduces both receptors with diminished clonogenicity and augmented phagocytosis. CRISPR-mediated CD47 and HER2 dual knockouts not only inhibit clonogenicity but also enhance macrophage-mediated attack. Dual antibody of both receptors synergizes with RT in control of syngeneic mouse breast tumor. These results provide the evidence that aggressive behavior of radioresistant BC is caused by CD47-mediated anti-phagocytosis conjugated with HER2-prompted proliferation. Dual blockade of CD47 and HER2 is suggested to eliminate resistant cancer cells in BC radiotherapy.

[1] Department of Radiation Oncology, University of California Davis, Sacramento, CA, USA. [2] Center for Molecular Medicine, Xiangya Hospital, Central South University, Changsha, Hunan, China. [3] Department of Surgery, School of Medicine, University of California Davis, Sacramento, CA 95817, USA. [4] Department of Pathology, Kaiser Permanente Medical Center Vallejo and Vacaville, Vallejo, CA, USA. [5] Department of Biochemistry and Molecular Medicine, University of California Davis, Sacramento, CA, USA. [6] NCI-Designated Comprehensive Cancer Center, University of California Davis, Sacramento, CA, USA. [7] Department of Pathology, Ohio State University, Columbus, OH, USA. [8] Department of Internal Medicine, University of California Davis, Sacramento, CA, USA. [9] Department of Radiation Oncology, Northwestern University, Feinberg School of Medicine, Chicago, IL, USA. [10] Department of Radiation and Cellular Oncology and the Ludwig Center for Metastasis Research, The University of Chicago, Chicago, IL, USA. [11] Chemoattractant Receptor and Signal Section, Cancer and Inflammation Program, Center for Cancer Research, National Cancer Institute, Frederick, MD, USA. [12] Department of Neurology, Johns Hopkins School of Medicine, Baltimore, MD 21205, USA. [13] Department of Dermatology, University of California Davis, Sacramento, CA, USA. [14]These authors contributed equally: Demet Candas-Green, Bowen Xie. ✉email: jijli@ucdavis.edu

Radiotherapy (RT) with the advantage of relatively less systemic side effects compared to chemotherapy is increasingly applied to treat breast cancer (BC) of all subtypes[1–3]. A meta-analysis of 17 randomized clinical trials of BC patients treated with RT after breast-conserving surgery demonstrated a reduction of 10-year risk in locoregional or distant tumors as well as of 15-year risk in BC death[4]. However, with the potential tumor adaptive radioresistance that increases recurrent and metastatic disease[1,5,6] and the prospective benefits of targeted immunotherapy (TI), RT combined with immunotherapy is being increasingly applied in the treatment of an array of human cancers[7–9]. Radiation-induced proteins and cytokines in the tumor and/or stroma cells including immune cells coordinatively contribute to the overall tumor response to treatment[10,11]. In addition to the well-defined PD-1 and PD-L1, a cluster of immune regulators are detected in the irradiated tumor microenvironment[9,12,13]. Thus, to further raise the synergetic efficacy of RT combined with TI, mechanisms of tumor-stromal communication[14,15], especially the dynamic communications between RT-surviving tumor cells and immune cells need to be investigated.

CD47, a myeloid-specific immune checkpoint protein[16] identified as a component of the Rh blood group antigen complex[17], is expressed in many human cancer cells[18]. CD47 on tumor cells can bind to its ligand SIRPα on macrophages[19], resulting in the phosphorylation of the cytoplasmic tail of SIRPα to initiate the signaling cascade for inhibiting the capacity of macrophage phagocytosis[20]. Cancer immunotherapy with humanized CD47 antibodies is developed[21] to augment the macrophage-mediated clearance on tumor cells[21,22], demonstrating an anti-tumor efficacy in human cancers[23] with tolerable toxicities[24]. However, in addition to enhancing tumor phagocytosis, blocking CD47 also improves the antigen presentation by DCs and inhibits the aggressive phenotype of breast cancer stem cells (BCSCs) via inhibition of EGFR signaling[25]. Such CD47-initiated proliferative events, especially its communication with growth factors in radioresistant cancer cells remains unclear.

This study reveals a coordinative transcriptional regulation of CD47 and HER2 in the radioresistant BC cells. Expression of both CD47 and HER2 is enhanced in irradiated BC cells, in RT-treated mouse syngeneic breast tumors, and in BC patients with recurrent diseases associated with a poor prognosis. Blocking CD47 not only enhances macrophages-mediated phagocytosis but also suppresses HER2-associated aggressiveness in the radioresistant BC cells, which is recapitulated using mouse syngeneic BC that can be synergistically suppressed by RT with inhibition of both receptors. These results demonstrate a latent pro-tumor growth mechanism due to the cross-talking of CD47-enhanced immune-avoidance with HER2-promoted intrinsic cell proliferation, causing the aggressive behavior of resistant BC cells. Dual blocking CD47 and HER2 may effectively abolish resistant cancer cells in BC radiotherapy.

## Results

**CD47 induction links to HER2 status**. CD47 and HER2, two well-defined cell surface receptors with critical biological functions, were found overexpressed in an array of human cancers including 1085 BC compared to the counterpart normal tissue via analysis of TCGA databases using the web-tool GEPIA (Fig. 1a and Supplementary Table 1). The potential functional conjugation between CD47 and HER2 was indicated by the striking difference in CD47-expression levels in HER2[+] versus HER2[−] BC cells (Fig. 1b), and HER2[+] versus HER2[−] tumors from BC patients (Fig. 1c, d). In addition, the elevated expression of both receptors was more frequently detected in the recurrent BC

tumors compared to the counterpart primary tumors by IHC analysis (Fig. 1e). Such CD47-HER2-associated recurrent rates also agreed with a worse prognosis in BC patients with a dual enhancement of CD47 and HER2 compared to patients with high expression of either receptor alone, measured by recurrence-free survival (RFS) or by distant metastasis-free survival (DMFS) (Fig. 1f, g); as well as by the overall survival (OS) in patients with lymph node metastasis (Supplementary Fig. 1a), or in patients with endocrine therapy after surgery (Supplementary Fig. 1b).

**CD47 and HER2 in radioresistant BC cells and irradiated tumors**. We hypothesized that the accumulated amount of CD47 and HER2 on the cell surface is a critical feature of an aggressive phenotype in radioresistant BC cells. The antiserum generated by immunization of mice with the radioresistant MCF7/C6 cells identified a cluster of immunogenic membrane proteins including CD47 and HER2 present in MCF7/C6 and in the cell membrane of radiation-derived breast cancer stem cells (RD-BCSCs)[26] (Supplementary Fig. 1c, d). Thus, co-expression of CD47 and HER2 in the radiation-surviving BC cells that contain stem-like cancer cells strongly suggests a latent two-layer pro-survival mechanism causing the aggressive behavior: CD47-enhanced capacity to escape immune surveillance coordinated with HER2-enhanced proliferative ability. Indeed, compared to the wild-type cancer cells, the radioresistant counterparts (MBA-MD-231/C5 and MCF7/C6) demonstrated strikingly elevated protein levels of CD47 and HER2 (Fig. 2a, b). The elevated expression of both receptors was also demonstrated in the syngeneic mouse orthotopic 4T1 breast tumors treated by local RT with single or fractionated ionizing radiation (FIR; Fig. 2c). In addition, increased cell populations co-expressing both receptors were confirmed with flow cytometry analysis (Fig. 2d and Supplementary Fig. 2a). The enhanced CD47 expression and cell population were further identified in the radioresistant MBA-MD-231/C5 and MCF7/C6 lines, both induced HER2 expression after FIR[26] (Supplementary Fig. 2b, c). Remarkably, compared to the lack of basal and induced CD47 in the normal human breast epithelial MCF10A cells that do not express HER2, CD47 expression was noticeably enhanced in BC and other human cancer cell lines even in the MCF7 cells in which a slightly elevated HER2 was detected (Fig. 2e) as well as in the counterpart xenograft tumors treated by in vivo radiation (Fig. 2f).

**CD47 transcription is regulated via HER2–NF-κB pathway**. Lapatinib, a small-molecule kinase inhibitor, has been reported to inhibit HER2, EGFR, and HER3[27,28], and approved for the treatment of advanced metastatic BC patients. HER2-mediated activation of PI3K-AKT pathway can cause NF-κB activation[29,30]. We have previously reported that HER2-mediated AKT activation stimulates NF-κB leading to the initiation of HER2 promoter[31]. Recently, HER2 is shown to recruit AKT to disrupt STING pathway causing immunosuppression to virus infection[32]. Such HER2-mediated immunosuppressive function is further evidenced by our current observation of HER2-induced CD47 upregulation in radioresistant BC cells. We found that blocking of HER2 by Lapatinib could efficiently reduce not only the CD47[+] population in MCF7/C6 that expresses HER2[26](Fig. 3a and Supplementary Fig. 3a) but also in HER2-expressing SKBR3 cells (Fig. 3b and Supplementary Fig. 3a, b). Such radiation-induced and HER2-dependant CD47 expression was further illustrated by a lack of CD47 enhancement in HER2-expressing RD-BCSCs treated by Herceptin (Fig. 3c). In addition, CD47[+] cell population was strikingly reduced in the HER2[+] RD-BCSCs with CRISPR-mediated knockout of HER2 (Fig. 3d and Supplementary Fig. 3c) and, remarkably, radiation could hardly induce CD47 expression

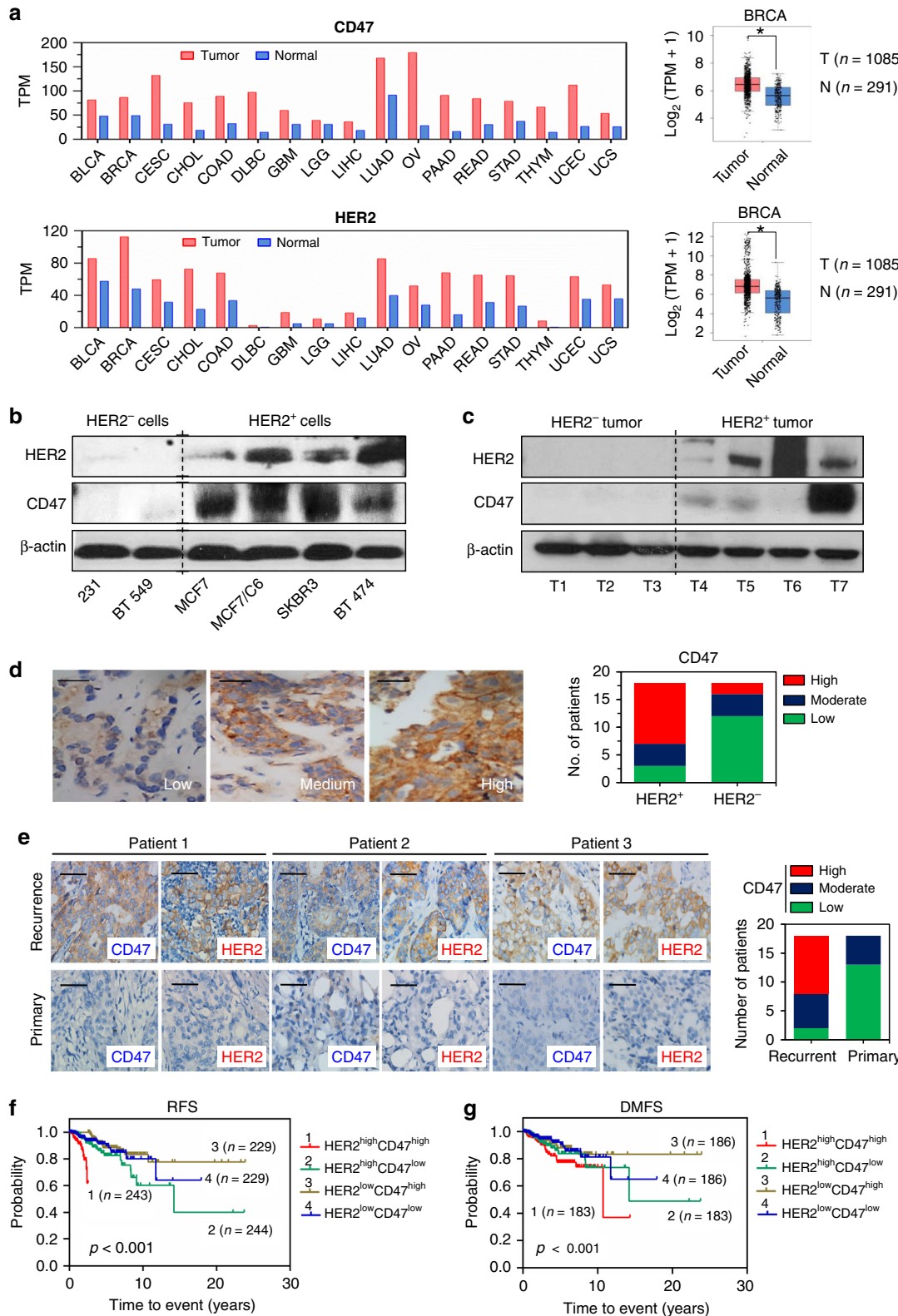

in the HER2$^{-/-}$ RD-BCSCs (Fig. 3e). As HER2 is shown to participate in radiation-induced NF-κB activation[29,31], these results clearly illustrate a possibility that radiation-induced CD47 transcription is regulated via HER2–NF-κB pathway.

We then used our identified NF-κB motif in the promoter of human *CD47* to test the CD47 transcription by Lapatinib (Supplementary Fig. 4a, b). Again, both of the basal and

radiation-induced CD47 promoter activity was remarkably suppressed in MCF7/C6 cells with CRISPR-mediated HER2 knockout (Fig. 3f). The HER2–NF-κB-controlled CD47 transcription was supported by the lack of promoter activity due to deficient NF-κB motif in the CD47 promoter stimulated with NF-κB activator TNF-α (Fig. 3g). It was further evidenced by a similar non-responsiveness to radiation in MCF7/C6 cells with

**Fig. 1 Expression of CD47 is linked with HER2 status. a** Co-expression of CD47 and HER2 across multiple human cancers (red) compared to corresponding normal tissues (blue) obtained from GEPIA database; expression of CD47 and HER2 in 1085 BC patients and 291 normal tissues are shown in the right. T tumors, N normal tissues. *$P < 0.05$. **b** Expression of CD47 preferring in HER2-expressing BC cells (MCF7, MCF7/C6, SKBR3, and BT474) compared to HER2-negative MDA-MB-231 and BT549 cells. **c** Immunoblotting of HER2 and CD47 in tumors from diagnosed HER2$^+$ and HER2$^-$BC patients. **d** Left, representative images scored as low, moderate, and high CD47 staining. Right, numbers of patients with low, medium, or high IHC staining of CD47 grouped by HER2 positive or negative status (total HER2$^+$ tumors $n = 18$; total HER2$^-$ tumors $n = 18$). **e** Left, representative IHC of CD47 and HER2 in paired primary and recurrent samples from the same patients. Right, numbers of patients with low, medium, or high IHC staining of CD47 grouped by primary or recurrent status (total primary tumors $n = 18$; total recurrent tumors $n = 18$). **f** Probability of recurrence-free (RFS) and **g** distant metastasis-free survival (DMFS) of BC patients of all subtypes stratified according to HER2 CD47 signature expression within HER2 strata from Breast Cancer Meta-base: 10 cohorts 22k genes database generated by SurvExpress (http://bioinformatica.mty.itesm.mx:8080/Biomatec/SurvivaX.jsp) from the HER2 probe 210930_s_at combined CD47 probe 211075_s_at. Statistical significance was analyzed by log-rank test.

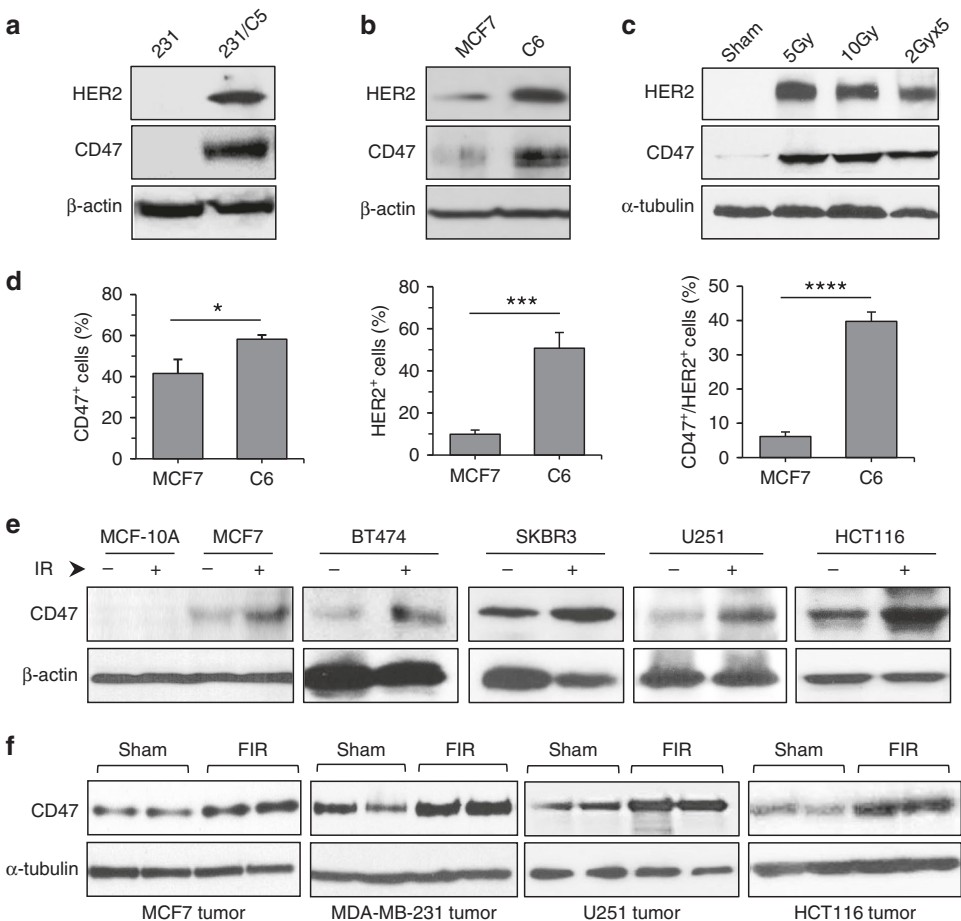

**Fig. 2 Induction of CD47 and HER2 in resistant BC cells and irradiated tumors.** Expressions of HER2 and CD47 in radioresistant MDA-MB-231/C5 cells (**a**) in radioresistant MCF7/C6 cells (**b**) both were derived from surviving clones after FIR. **c** Expression of HER2 and CD47 in syngeneic mouse 4T1 breast tumors 16 h after local RT with single-dose radiation or FIR (pooled samples from six tumors). **d** Enhanced CD47$^+$/HER2$^+$ population in MCF7/C6 cells compared to parental MCF7 cells measured by flow cytometry ($n = 3$; *$P < 0.05$, ***$P < 0.001$, ****$P < 0.0001$). **e** Induction of CD47 by radiation in an array of human cancer cells (breast cancer MCF-7, BT474, SKBR3, glioblastoma U251, and colorectal carcinoma HCT116 cells) measured by immunoblotting 16 h after radiation (5 Gy). **f** Induction of CD47 expression in xenograft tumors (duplicated samples from each tumor group) following RT with fractionated doses (FIR, 2 Gy × 5).

the NF-κB-deficient promoter or treated with NF-κB inhibitors (IMD-0354, MLN120B, BMS-345541) or Herceptin (Fig. 3h and Supplementary Fig. 4c, d). Strikingly, NF-κB inhibitors (IMD-0354, MLN120B, BMS-34554) suppressed the radiation-induced CD47 expression in MCF7 or/and MCF7/C6 cells (Fig. 3i, j and Supplementary Fig. 4e, 4f). Using the IκB promoter as a positive control for NF-κB DNA binding capacity, a ChIP assay identified the recruitment of NF-κB/p65 to the CD47 promoter in MCF7 cells treated with radiation

or TNF-α (Fig. 3k). Moreover, in the light of a recent report of CD47 blockade enhancing trastuzumab efficacy on HER2 positive BC[33], the percentage of HER2-expressing cells was found to be substantially reduced from 20.2% in the control MCF7/C6 cells to 3.66% in the MCF7/C6 cells with CRISPR-mediated CD47 deficiency (Fig. 3l). Collectively, these results demonstrate that CD47 and HER2 are coordinately upregulated at a transcriptional level causing activation of two receptors in the radioresistant BC cells.

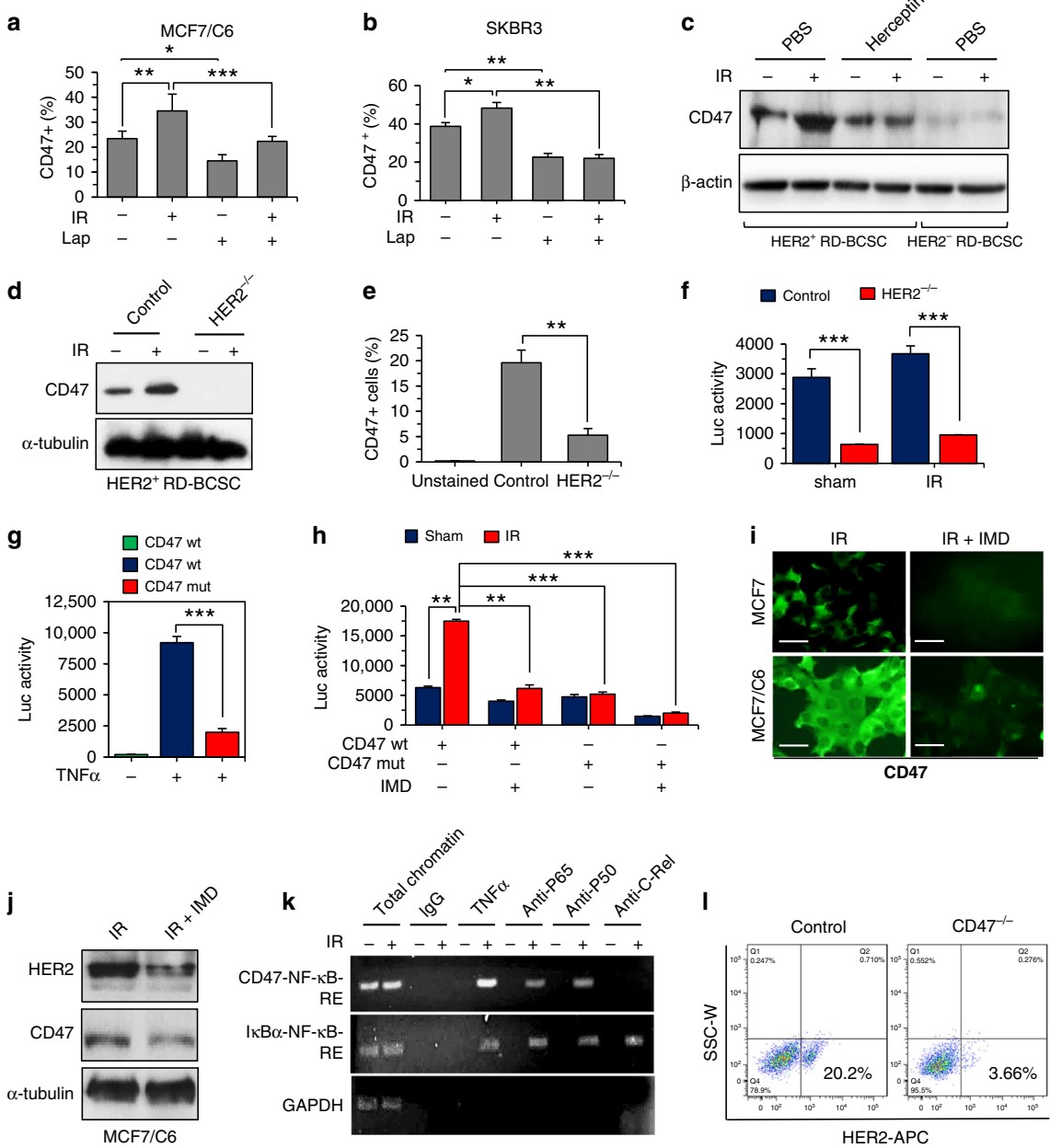

**Fig. 3 Radiation-induced CD47 transcription is regulated via HER2–NF-κB axis.** Radiation enhanced CD47+ cell populations were reduced by inhibition of HER2 in MCF7/C6 (**a**) and in SKBR3 cells (**b**) treated with Lapatinib (10 μM, 72 h) in the presence or absence of IR (5 Gy) (n = 3; *P < 0.05, **P < 0.01, ***P < 0.001). **c** Immunoblotting of CD47 in HER2+ RD-BCSCs treated with PBS or Herceptin (10 μg/ml for 5 days refreshed on 2nd and 4th day) followed by sham or 5 Gy IR (HER2−RD-BCSCs included as a control)· **d** CD47+ populations detected by flow cytometry in HER2+ RD-BCSCs and HER2+ RD-BCSCs with CRISPR-mediated knockout of HER2 (HER2−/−), n = 3; **P < 0.01. **e** Basal and radiation-induced CD47 expression was absent in HER2+ RD-BCSCs with CRISPR-mediated knockout of HER2 (HER2−/−). **f** NF-κB luciferase reporter activity in radioresistant MCF/C6 cells and MCF/C6 cells with HER2−/−, n = 3, ***P < 0.001. **g** CD47 promoter-controlled luciferase reporter activity containing wild-type or mutant NF-κB-binding motif in MCF7/C6 cells treated with TNFα (n = 3, ***P < 0.001). **h** CD47 promoter activity with wild-type or mutant NF-κB-binding motif measured in control and irradiated MCF7/C6 cells in the presence or absence of NF-κB inhibitor IMD-0354 (2 μM, 5 h; n = 3, **P < 0.01, ***P < 0.001). **i** Representative images of immunofluorescence of CD47 in irradiated MCF7 and MCF7/C6 cells pretreated with IMD-0354 (IMD); CD47 was visualized by confocal microscope 16 h after IR. Scale bar, 25 μm. **j** Immunoblotting of CD47 and HER2 in IR-treated MCF7/C6 cells with or without IMD-0354. **k** ChIP-qPCR assay for NF-κB recruitment in human CD47 promoter in MCF7/C6 cells irradiated (5 Gy) or treated with 10 ng/ml TNF-α. Chromatin precipitation was conducted with anti-p65 where anti-c-Rel served as a positive control in IκB-α promoter, IgG served as a negative control. **l** Reduced HER2-expressing populations in MCF7/C6 cells with CRISPR/Cas9-mediated knockout of CD47.

**CD47 and HER2 crosstalk in phagocytosis and clonogenicity.** With the defined tumor-promoting functions of tumor-infiltrating macrophages[34], we speculate that the lack of tumor attack by macrophages in radioresistant cancer cells is at least in part due to the enhanced immune-defending capacity by HER2–NF-κB-mediated CD47 enrichment. DDAO-labeled MCF7 and MCF7/C6 cells were co-cultured with the mature human macrophages derived from the THP1 cells and anti-CD47 antibody (produced in B6H12.2 hybridoma). Indeed, the radio-resistant MCF7/C6 cells showed a remarkably reduction of

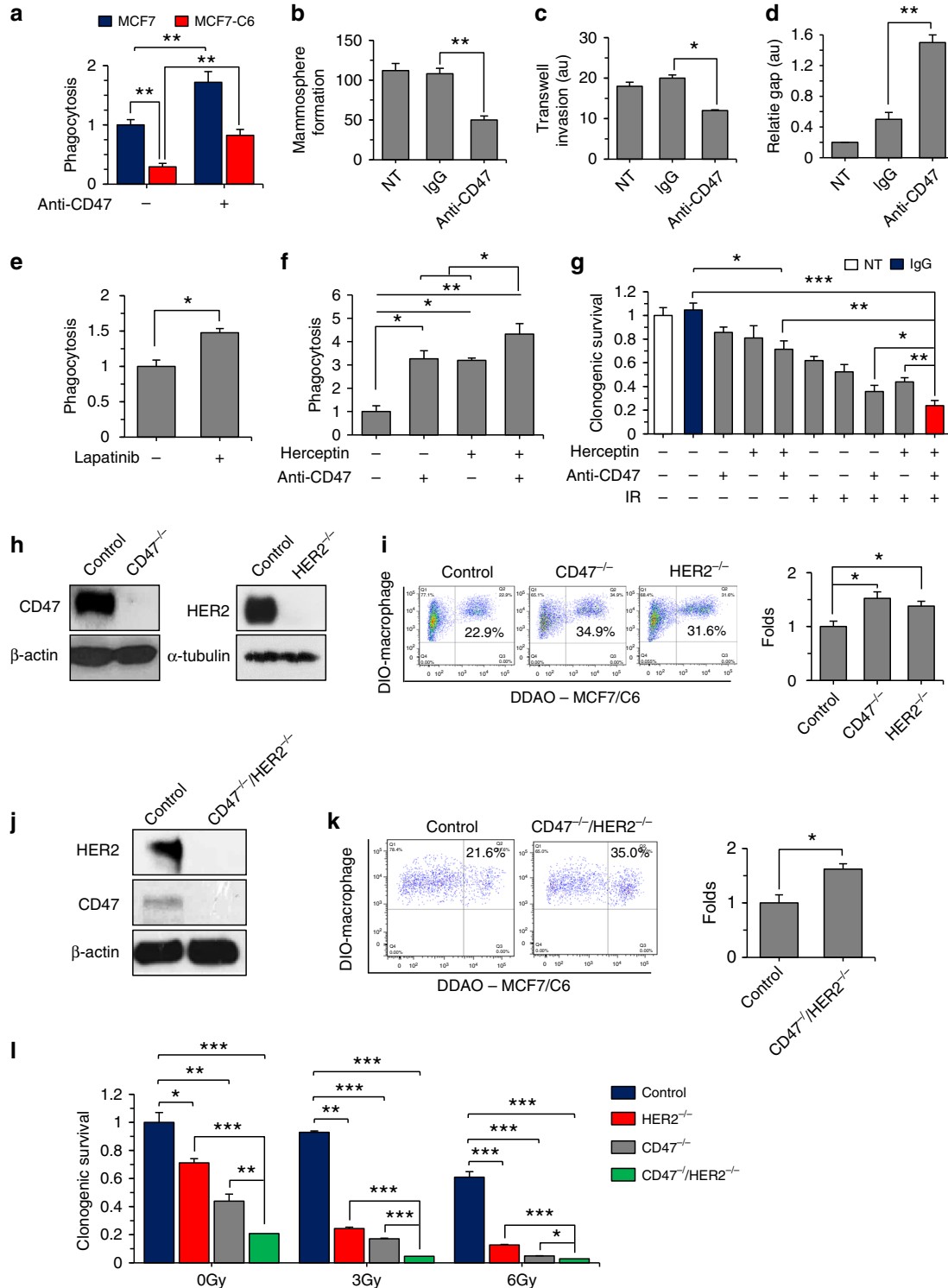

phagocytosis compared to the MCF7 cells, and in agreement with literature[35], the CD47 antibody significantly promoted macrophage-mediated phagocytosis on both MCF7 and radio-resistant MCF7/C6 cells (Fig. 4a and Supplementary Fig. 5a, b), indicating that CD47 is an effective immune checkpoint target to synergize RT in BC treatment. However, with above-demonstrated cross-regulation between CD47 and HER2, a leakage of CD47 due to HER2 expression could compromise such synergy. Thus, we further tested the potential crosstalk of these two receptors in signaling the aggressive growth of MCF7/C6 cells containing RD-BCSCs that mimic the heterogenic feature of a

solid BC. The aggressive growth features including mammosphere formation (Fig. 4b and Supplementary Fig. 5c), matrigel invasion (Fig. 4c and Supplementary Fig. 5d), and gap-filling capabilities (Fig. 4d and Supplementary Fig. 5e) were considerably suppressed by treatment with the anti-CD47 antibody. Intriguingly, the anti-CD47 inhibited aggressive growth is associated with the HER2 pathway since macrophage-mediated phagocytosis on MCF7/C6 cells can be enhanced by Lapatinib (Fig. 4e and Supplementary Fig. 6a, b). Impressively, although single antibody of either receptor can induce phagocytosis, dual antibody showed an obvious synergy in the macrophage

**Fig. 4 CD47/HER2 blockage maximizes the clonogenic radiosensitivity and macrophage phagocytosis. a** Macrophage-mediated phagocytosis was enhanced in both MCF7 and MCF7/C6 cells by CD47 antibody detected by flow cytometry analysis with THP1-derived mature macrophages stained with DIO and incubated with $3 \times 10^4$ DDAO-labeled tumor cells at a ratio of 5:1 for 2 h at 37 °C. For anti-CD47 treatment, tumor cells were pre-incubated with 10 µg/ml of CD47 antibody (B6H12. 2) for 1 h prior to incubation with macrophages ($n = 3$, **$P < 0.01$). Mammosphere formation (**b**, $n = 3$; **$P < 0.01$), transwell invasion (**c**, au arbitrary units; $n = 3$; *$P < 0.05$), and gap-filling assay (**d**, $n = 3$; **$P < 0.01$) of MCF7/C6 cells with or without anti-CD47 treatment (no treatment and IgG were used as controls). **e, f** Enhanced macrophage-mediated phagocytosis on MCF7/C6 cells treated with Lapatinib, Herceptin, B6H12.2, or combination of Herceptin and B6H12. 2 in MCF7/C6 cells ($n = 3$, *$P < 0.05$, **$P < 0.01$). **g** Clonogenic survival of MCF7/C6 cells received radiation with different antibody combinations (IgG or B6H12. 2, 10 µg/ml for overnight; Herceptin, 10 µM for 72 h; IR, 5 Gy) was calculated by normalizing to the clonogenic fractions of control cells received 0 Gy IR ($n = 3$, *$P < 0.05$, **$P < 0.01$). **h** Lack of targeted gene expression in $CD47^{-/-}$ or $HER2^{-/-}$ in MCF7/C6 cells. **i** Enhanced macrophage phagocytosis on $CD47^{-/-}$ or $HER2^{-/-}$ MCF7/C6 cells (*$P < 0.05$). **j** Immunoblotting of CD47 and HER2 in MCF7/C6 cells with $CD47^{-/-}/HER2^{-/-}$ status. **k** Macrophage phagocytosis of $CD47^{-/-}/HER2^{-/-}$ MCF7/C6 cells (*$P < 0.05$). **l** Clonogenic radiosensitivity of MCF7/C6 cells with $CD47^{-/-}$, $HER2^{-/-}$, or $CD47^{-/-}/HER2^{-/-}$ status calculated by normalizing to the clonogenic fractions of control cells received 0 Gy IR ($n = 3$, *$P < 0.05$, **$P < 0.01$, ***$P < 0.001$).

phagocytosis of the radioresistant BC cells (Fig. 4f and Supplementary Fig. 6c), indicating that HER2–NF-κB-mediated CD47 transcriptional activation is responsible for the potential tumor resistance to anti-CD47 immunotherapy. In support of this concept, cell clonogenicity of MCF7/C6 cells were mostly inhibited by radiation combined with dual blockade of CD47 and HER2 by antibodies (Fig. 4g and Supplementary Fig. 6d). However, CRISPR-mediated deficiency of CD47 or HER2 (Fig. 4h, i) or dual deletion of CD47/HER2 (Fig. 4j, k) revealed an unexpected result. The macrophage phagocytosis was similarly enhanced in MCF7/C6 cells with $CD47^{-/-}$ or $HER2^{-/-}$ and no further increased phagocytosis was detected in the dual-deficient $CD47^{-/-}/HER2^{-/-}$ cells, whereas clonogenicity was suppressed by dose-dependent radiosensitization with double gene deficiency compared to single gene deficiency (Fig. 4l). Thus, these results support the finding that protein crosstalk in the cytoplasm is required for the mutually gene activation of the two receptors; whereas $CD47^{-/-}$ or $HER2^{-/-}$ cells no longer have such gene transcription regulation due to lack of the protein crosstalk. Importantly, the antibody-targeted single surface receptor may not be able to block the potential communication in the cytoplasm such as currently identified crosstalk between CD47 and HER2 for which gene knockout may hold the latent advantage. Therefore, due to their potential crosstalk under the DNA-damaging anticancer modalities including RT that may otherwise activate immune attack by inducing tumor immunogenicity[36]. Therefore, blocking of either receptor may not be sufficient to totally eliminate the radioresistant cancer cells.

**Radiosensitization of $CD47^{-/-}/HER2^{-/-}$ tumors**. To further elucidate the potential synergies of RT and dual CD47 and HER2 blockade, a radioresistant mouse breast tumor 4T1/C2 cell line was generated following the same protocol used for generating MCF7/C6 cells[26]. The radioresistant 4T1/C2 cells were then genetically edited by CRISPR technology to establish cells with $CD47^{-/-}$, $HER2^{-/-}$, or $CD47^{-/-}/HER2^{-/-}$ status that were then used to create the syngeneic mouse orthotopic breast tumors. In consistent with the in vitro studies, the $CD47^{-/-}/HER2^{-/-}$ tumors showed a further limited growth although deletion of either receptor also decreased growth to a certain degree (Fig. 5a, b). Local breast tumor radiation with a single dose 5 Gy delivered on day 9 further synergized the inhibition on $CD47^{-/-}/HER2^{-/-}$ tumors, although again, the $CD47^{-/-}$ and $HER2^{-/-}$ tumors also generated an intermediate response compared to radiation alone (Fig. 5c, d). Agreed with the tumor growth inhibition, the tumor-infiltrated macrophages detected with CD11b staining were more enhanced in the irradiated $CD47^{-/-}/HER2^{-/-}$ tumors compared to tumors with single-gene deletion (Fig. 5e), which was correspondingly associated with an increased necrosis detected in the $CD47^{-/-}/HER2^{-/-}$

tumors compared to the $CD47^{-/-}$ or $HER2^{-/-}$ tumors (Supplementary Fig. 7). These results demonstrate a potential therapeutic approach using CRISPR-mediated dual deficiency of CD47 and HER2 in BC radiotherapy.

**Radiosensitization synergized with dual receptor inhibition.** With the encouraging evidences of efficacious anti-CD47 immunotherapy[33,37] and radiation-enhanced anti-CD47-mediated phagocytosis (Supplementary Fig. 8), the potential synergy betwen radiation and antibody-mediated CD47 blockade was tested using the syngeneic mouse orthotopic BC model. Interestingly, anti-CD47 treatment was maximized when started day 1; whereas a slight synergistic tumor inhibition was observed with radiation combined with an anti-CD47 antibody that indeed increased tumor-infiltrated macrophages (Supplementary Fig. 9a, b, c). Repeated tests using the syngeneic mouse orthotopic BC model treated with RT (5 Gy × 2) and antibodies of CD47 and HER2 (no detectable toxicity measured by body weight; Supplementary Fig. 10a), demonstrate an enhanced synergistic tumor inhibition by RT with the dual receptor blockade (Fig. 6a–c) with the correspondingly increased necrotic area and infiltrated macrophages (Fig. 6d), although control tests without radiation (Supplementary Fig. 10b, c, d) or RT combined with either antibody of CD47 or HER2 (Fig. 6a–c), also showed a certain level of synergistic tumor growth. Importantly, both the tumor-infiltrated macrophages and macrophage-mediated phagocytosis were increased in the tumors treated by RT combined with dual antibodies, although a combination of RT with single antibody also increased a significant amount of the infiltrated macrophages and phagocytosis in the orthotopic GFP-4T1 breast tumors (Fig. 6e–g). It is worthy to note that tumors treated with RT and Herceptin showed the enhanced amounts of macrophages and macrophage-mediated phagocytosis although lowered than Herceptin induced phagocytosis compared to CD47 antibody detected by in vitro analysis shown in Fig. 4f. Again, these findings support active crosstalk between CD47 and HER2 in a radiation-treated tumor microenvironment. Altogether, these results suggest that CD47 and HER2 are mutually activated at the transcriptional level contributing to the aggressive phenotype of radiation-resistant breast cancer cells via HER2–NF-κB-controlled CD47 promoter activation (Fig. 7). Thus, HER2-expressing tumor cells including the HER2-expressing cancer stem cells[26] may take advantage of CD47-mediated immunosuppression to not only survive radiation but also escape immunosurveillance, contributing to the aggressive phenotype of radioresistant BC. A dual blockade of these two receptors is a potentially effective strategy to eliminate radioresistant BC cells.

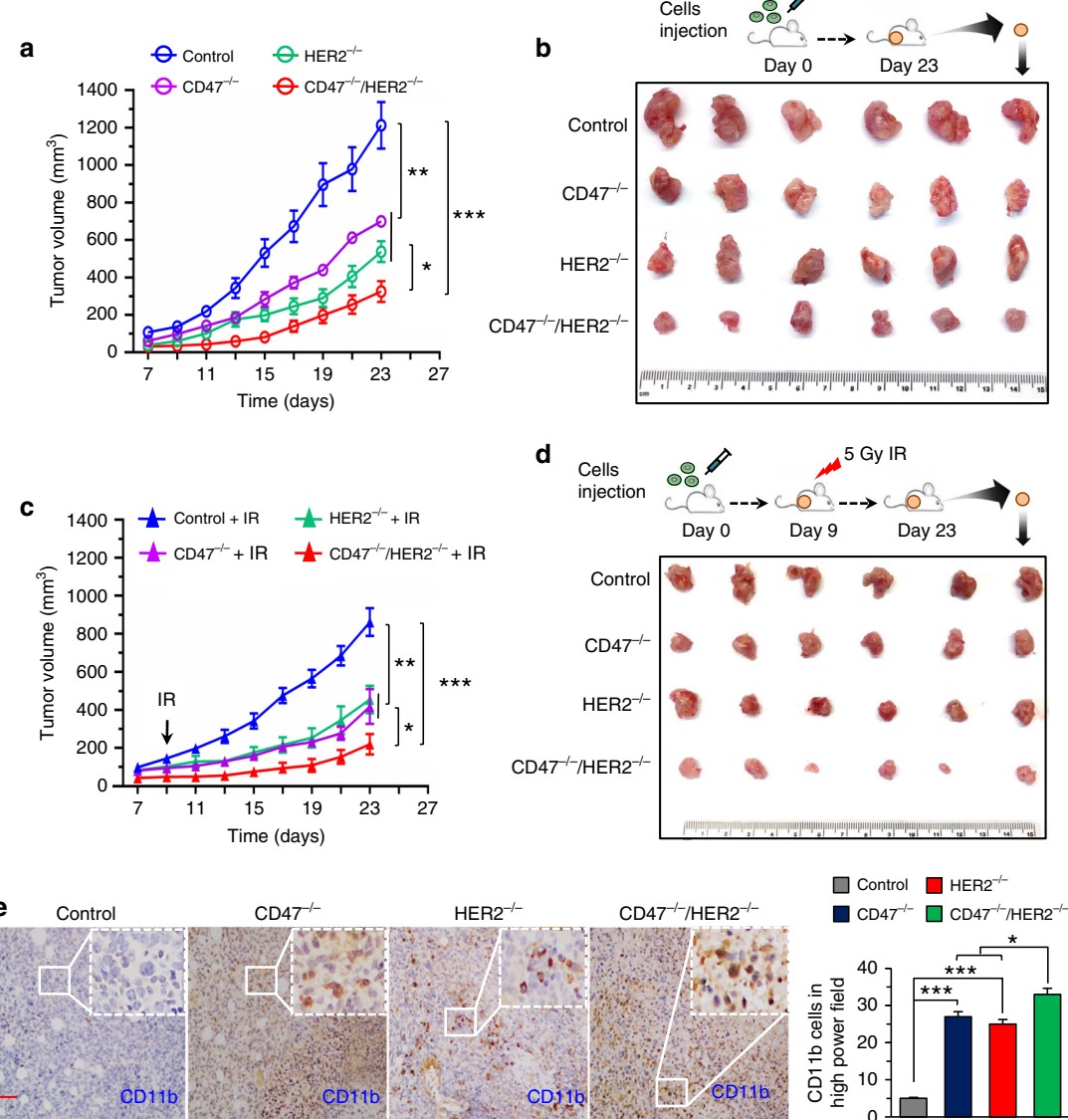

**Fig. 5 Radiosensitization is maximized in mouse breast tumors carrying CD47$^{-/-}$/HER2$^{-/-}$ status. a** Growth rate of mouse breast tumors generated with radioresistant 4T1/C2 cells with CD47$^{-/-}$, HER2$^{-/-}$, or CD47$^{-/-}$/HER2$^{-/-}$. **b** Representative images of tumors at the end of experiments. **c** Repeated experiment of **a** with a single dose IR (5 Gy) at day 9 and representative images of the tumor at the end of experiments shown in **d**; in **a** and **c**, n = 6, mean ± SEM, *P < 0.05, **P < 0.01, ***P < 0.001. **e** Left, representative IHC images of CD11b$^+$ macrophages in CD47$^{-/-}$, HER2$^{-/-}$, CD47$^{-/-}$/HER2$^{-/-}$ tumors; scale bar = 100 μm; right, quantitation of CD11b$^+$ macrophages (n = 4, *P < 0.05, ***P < 0.001).

## Discussion

Increasing clinical trials are encouraged to evaluate the synergistic benefits in cancer control by radiotherapy combined with differently targeted immunotherapy[38], indicating a need for further elucidating the molecular insights underlying the interplay between tumor cells and immune cells in the radiation-treated tumor microenvironment. Currently, little evidence is available on whether radiation-induced tumor adaptive response could be overlapping with the acquired immune tolerance in cancer immunotherapy[39]. Thus, to further enhance the synergistic tumor inhibition by RT and immunotherapy, in addition to further elucidating radiation-associated tumor immunogenicity, radiation-inducible immunosuppressive factors, especially on the radioresistant cancer cells, need to be identified. This study provides the evidence revealing that two receptors with different biological functions are coordinately contributing to the aggressive phenotype of radioresistant BC cells. The intrinsic growth

potential by HER2 is conjugated with CD47-mediated extrinsic anti-phagocytosis, demonstrating a need of an integrated therapeutic approach to target multiple factors to eliminate resistant cancer cells in the combined modality of RT with immunotherapy.

The synergistic tumor inhibition of RT combined with immunotherapy is attributed to the well-defined radiation-associated tumor priming function (abscopal effect)[40,41] which has been well-evidenced by induced cell surface epitopes processed and presented in radiation-treated tumors[7], enhanced diversity of the T-cell receptor repertoire of intratumoral T cells[42], synergic potentiation in improving immune repertoire[43,44], and inhibited metastasis by RT with immunotherapy[45]. However, such radiation-associated anti-tumor immune response is rarely detected in clinic[46], implicating a possibility that radiation-induced immunosuppressive factors may compromise the anti-tumor immune regulation induced by radiation-associated tumor immunogenicity. Thus,

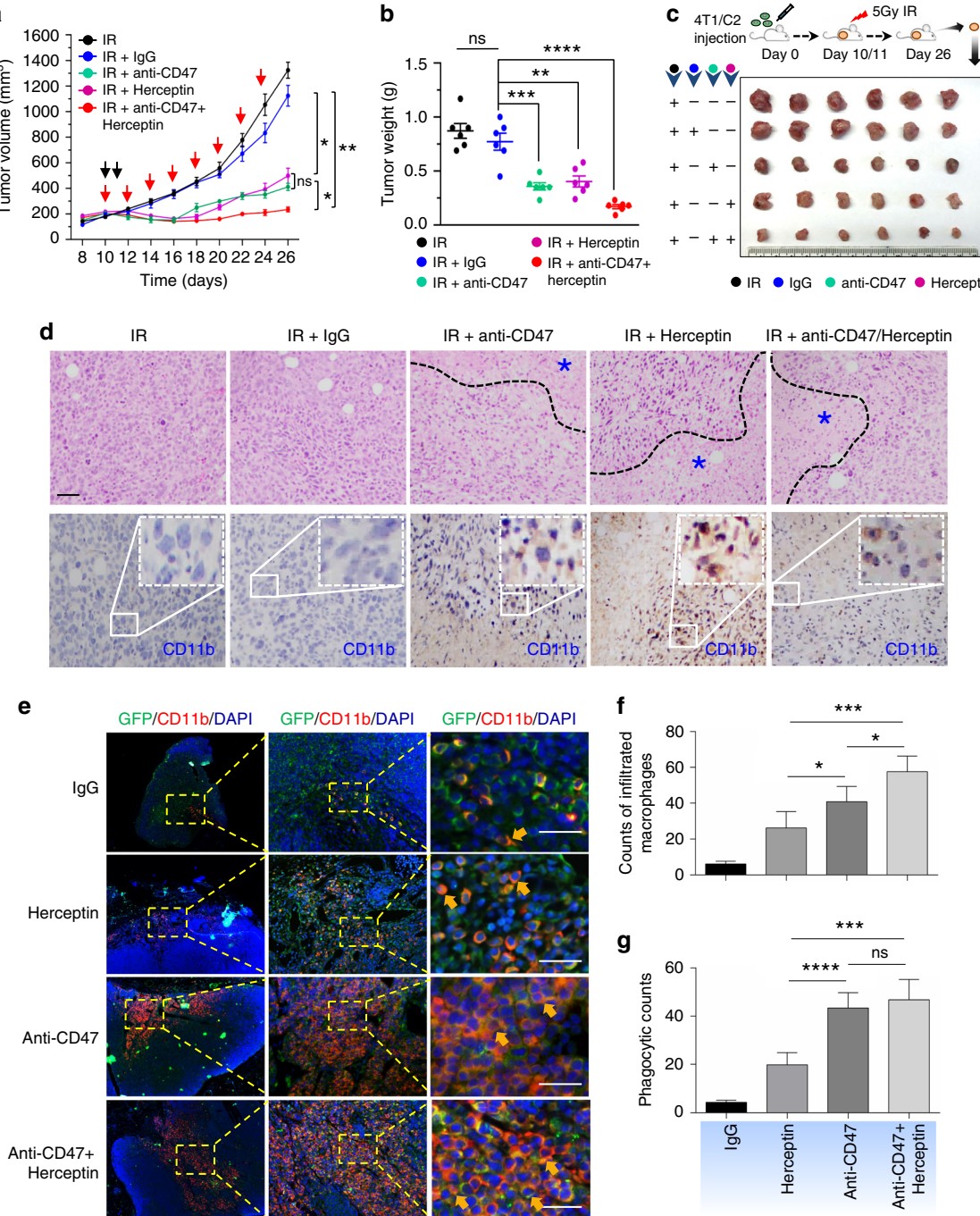

**Fig. 6 The synergetic tumor inhibition is enhanced by RT with dual antibody blockage of CD47 and HER2.** Tumor volumes (**a**), tumor weights at end of experiments (**b**), and representative images (**c**) were measured with mouse syngeneic breast tumors generated with radioresistant 4T1/C2 cells and treated with intratumoral injection of PBS, IgG, Herceptin (5 mg/kg), anti-CD47 antibody (5 mg/kg), or a combination both antibodies every other day starting at day 10 with radiation delivered locally to tumors of all groups with 5 Gy at days 10 and 11 (*n* = 6, mean ± SEM, ns not significant, *P < 0.05, **P < 0.01, ***P < 0.001, ****P < 0.0001). **d** Above, HE staining of 4T1/C2 tumors treated with radiation combined with indicated antibodies. Bottom, representative IHC images of CD11b⁺ macrophages in tumors; scale bar = 100 μm. **e** Representative images of infiltrated macrophage and macrophage-mediated phagocytosis in the syngeneic mouse orthotopic BC models repeated as in **a** with GFP-labeled 4T1 tumors detected with indicated antibodies; green, tumor cells; red, infiltrated macrophages; blue, cell nucleus stained with DAPI; yellow (indicated with arrows), macrophage-mediated phagocytosis with tumor cells. **f** Quantitation of relative richness of infiltrated CD11b⁺ macrophages. **g** Quantitation of macrophages-mediated phagocytosis with tumor cells counted in multiple areas in the high-power fields marked with yellow arrows (Image Pro Plus 6.0; **b**, **c**, *P < 0.05, ****P < 0.0001; ns not significant).

CD47 expression demonstrated in the radiation-surviving HER2-expressing breast cancer cells supports the concept that both anti- and pro-tumor immune factors can be enhanced by tumor cells treated by radiotherapy. Multiple factors with varied cellular functions such as HER2 and CD47 are associated with the tumor acquired resistance.

These results shed a light on the potential mechanistic crosstalk between cell growth factor and immunosuppressing receptor at

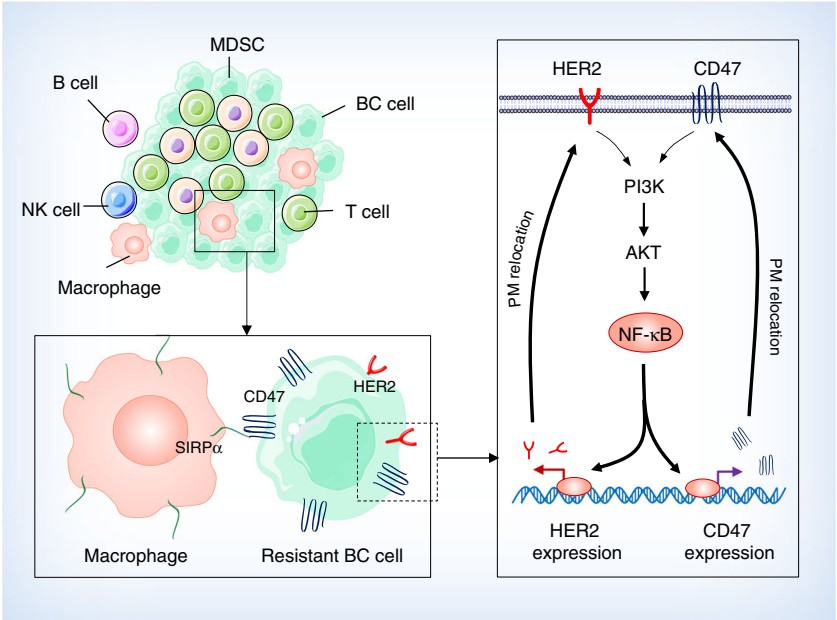

**Fig. 7 A cellular model of radioresistant breast cancer is proposed in which two cell surface receptors CD47 and HER2 coordinately contribute to the aggressive phenotype of breast cancer cells surviving radiotherapy.** The radioresistant breast cancer cells that are enriched with breast cancer stem cells can not only survive radiation but also enhance their immune-escaping ability via CD47-mediated anti-phagocytosis. CD47 transcription is regulated via HER2–NF-κB axis leading to the enhancement of both receptors on the cell surface. Thus, HER2-promoted intrinsic proliferation is synergized with the CD47-initiated extrinsic anti-phagocytosis; a dual blockade of these two receptors may eliminate the radioresistant breast cancer cells in breast cancer radiotherapy.

the transcriptional control. The aggressive behavior of radio-resistant BC cells is identified to be caused by two cell surface receptors via the transcription factor NF-κB as shown in Fig. 7. NF-κB, a well-defined stress-responsible transcriptional factor, plays a key role in gene regulation of both tumor cells and stromal cells including immune cells[47]. Thus, it is not surprising to find that NF-κB is involved in the regulation of both pro-tumor immune protection and radioresistance. In addition, it may be also coordinated with the pro-tumor function of NF-κB in tumor-associated macrophages[48]. The HER2–NF-κB axis[49] which is responsible for the co-expression of CD47 and HER2 transactivation that is able to generate a double layer pro-survival network for the radioresistant cancer cells. The enhanced expression of CD47 in the irradiated tumor microenvironment could severely hamper tumor radiosensitivity[50,51]. Elevated CD47 expression is reported in HER2 positive breast cancer HCC1954 and MCF7 cells and NF-κB and PPARα are detected in the super-enhancer of CD47[52]. The association between CD47 and HER2 may also explain the acquired resistance in BC patients treated with anti-HER2 modality and the limited tumor inhibition with RT combined with CD47 blockade. Agreed with the reported partial tumor growth inhibition by RT combined with blocking CD47[50,53], a similar insufficiency in tumor growth inhibition was observed by RT combined with an antibody blocking either CD47 or HER2 or with CRISPR-mediated gene deficiency. More recently CD47 was found to have another co-expression model in breast cancer in which the glucose-regulated protein 78 (GRP78) was linked with CD47 function[54]. Thus, a strategy targeting multiple membrane receptors should be considered in the combined therapy of RT and immunotherapy for recurrent and metastatic tumors.

Synergetic benefits of targeting multiple receptors are being evidenced including simultaneous blocking of CD47 and PD-L1 in cancer immune responses and cytokine release[55] and dual antibody blockade of CD39 and CD73 combined with ICIs and chemotherapies[56]. CD11b activation is required for macrophage activation and tumor cell phagocytosis[57]. Our finding that infiltrated CD11b+ macrophage and its phagocytosis on tumor cells were more enhanced in the dual inhibition of CD47 and HER2 and anti-CD47 inhibited HER2-mediated proliferation which is agreed with the finding that knockdown or antibody blockade of CD47 or SIRPα is effective in eliminating trastuzumab-adaptive HER2-expressing BC cells[58].

It has been shown that both innate and adaptive immunity are involved in anti-HER2 therapeutic efficacy[59]. Identification of the functional crosstalk of CD47 and HER2 reaffirms the potential benefit by RT with dual blocking CD47 and HER2 due to the fact that the immune reaction is mainly activated in the locally irradiated tumors with limited side effects to normal tissues[53]. Interestingly, CD47 expression is reported to be reduced by radiation in human papillomavirus-positive cancer resulting in immune-mediated clearance of tumor cells[60]. Thus, it is important to determine whether HER2 could be silenced or not be induced by radiation in papillomavirus-expressing cancer cells. In addition, a recent report further revealed that tumor intrinsic signaling of CD47 is able to regulate glycolysis and promotes colorectal cancer cell growth and metastasis[37], further supporting our finding that CD47-HER2 crosstalk is actively involved in tumor metastatic potential. Using the mouse syngeneic mouse breast tumors with CD47$^{-/-}$/HER2$^{-/-}$ status, we demonstrate an enhanced tumor inhibition by radiation combined with dual CD47 and HER2. Antibody-mediated dual inhibition of CD47 and HER2 confirms the advantage of local irradiation compared to inhibition of either receptor alone, although substantial tumor inhibition was also achieved by radiation combined with antibody blockage of CD47 or HER2. In addition to the well-defined ICIs including PD-1/PD-L1 and CD47, other radiation-responsive immunosuppressing, as well as immunostimulatory factors in the

tumor microenvironment, need to be elucidated for further enhancing the synergetic benefits for the combined anticancer modalities by radiation and immunotherapy. Further identification of the cell membrane proteins coordinatively induced or enhanced on tumor cells and stromal cells with immune regulatory functions will provide further mechanistic insights into the immune response in the tumor microenvironment under radiotherapy[9,12,13]. By antiserum analysis, CD47 and HER2 together with other polypeptides were shown to be accumulated on the cell surface in radioresistant MCF7/C6 and RD-BCSCs. The potential association of CD47 and HER2 with this cluster of membrane factors may contribute to the pro-tumor growth which may also compromise the radiation-induced anti-tumor immune regulation[61,62].

In summary, this study demonstrates that CD47 and HER2 functioning respectively in extrinsic immunosuppression and intrinsic proliferation, can coordinate contributing to the aggressive behavior of radioresistant BC cells. Cancer cells with HER2-promoted proliferation may be equipped with an additional survival advantage due to CD47-mediated anti-phagocytosis and vice versa, leading to an immunosuppressive tumor microenvironment and a lack of treatment response. Thus, for the treatment of recurrent tumors expressing HER2, a dual blockage of CD47 and HER2 may lead to the synergetic tumor inhibition by radiotherapy.

## Methods

**Cell lines**. Human breast cancer MCF-7, MDA-MB-231, BT474, BT549, SKBR3 cell lines, glioblastoma U251, and colon cancer HCT116 cell lines were purchased from ATCC. MCF-7, MDA-MB-231, BT474, BT549, and U251 cells were maintained in DMEM medium supplemented with 10% FBS, and SKBR3 cells maintained in RPMI-1640 medium with 10% FBS. HCT116 cells were maintained in McCoy's 5a Medium with 10% FBS. The radioresistant MCF7/C6 and MDA-MB-231/C5 cell lines that were cloned from surviving fractions of MCF7 and MDA-MB-231 cells after repeated irradiation; the HER2-overexpressing breast cancer stem cells (HER2$^+$/CD44$^+$/CD24$^{-/low}$ BCSCs) were isolated from MCF7/C6[26]. Mouse triple-negative breast cancer 4T1 cells were maintained in DMEM medium with 10% FBS. Human monocyte THP-1 were cultured in ATCC-formulated RPMI-1640 medium supplemented with 10% FBS, 15 mM HEPES, 4.5 g/l glucose, and 2-mercaptoethanol to a final concentration of 0.05 mM. Mouse Mφ RAW 264.7 cells were cultured in DMEM medium with 10% FBS following with the ATCC culture method. All cell lines were tested negative mycoplasma contamination with MycoSensor PCR Assay kit (Agilent Technologies, Catalog # 302108).

**Plasmids and reagents**. CD47 promoter-controlled luciferase vector was constructed by cloning the human CD47 promoter region (−1554 nt) into pGL2-basic plasmid from KpnI/Hind III sites. The deletion of the putative NF-κB-binding sites (TGGAAGCT-764-757) was performed using a quick mutation kit (Stratagene, La Jolla, CA). The primer sequences used: 5′-GTGGTCGGGTACCTGCCCGCTCGC CCCTCGCGGGCCTCTGCG-3′ (sense) and 5′-CGCAGAGCCCGCGAGGGGCGA GCGGGCAGGTACCCGACCAC-3′ (antisense).

pGL2-NF-κB-luc plasmid was constructed with the 3 NF-κB-binding sites: GTGGGTTCCC, TCGCCACTCCCC, and TGGAAAGTCCCC by blunt-end ligation. IMD-0354 (Catalog # I3159) and TNF-α (Catalog # T0157) were purchased from Sigma. The lentiCRISPR v2 vector was purchased from the Addgene plasmid repository (Catalog # 52961). Herceptin was obtained from the UC Davis Comprehensive Cancer Center Pharmacy as left-over medicine (Genentech, NDC 50242-132-01). Lapatinib (Catalog # S1028) was purchased from Selleckchem. Anti-CD47 (B6H12) antibody for IHC, ICC, and western blot was purchased from Santa Cruz (Catalog # sc-12730). Anti-CD47-FITC for flow cytometry was purchased from BD Biosciences (Catalog # 556045). Anti-human CD47 (B6H12) antibody for phagocytosis assay was extracted from B6H12.2 hybridoma (ATCC® HB-9771™). Anti-mouse CD47 antibody for in vivo tumor inhibition was extracted from MIAP301 hybridoma provided by Dr. William Frazier (Washington University School of Medicine). Anti-CD11b antibody for IHC staining was purchased from Invitrogen (Catalog # MA5-17857). Mouse monoclonal anti-α-tubulin antibody for western blot was purchased from Sigma-Aldrich (Catalog # T6074). Mouse monoclonal anti-β-actin antibody for western blot was purchased from Sigma-Aldrich (Catalog # A5441). Anti-HER2/ERBB2 (Catalog # 29D8) antibody for IHC staining and western blot was purchased from Cell Signaling Technology (Catalog # 2165). Anti-HER2-APC antibody for flow cytometry analysis was purchased from BD Biosciences (Catalog # 340554).

**Immunoblotting**. Proteins from cells or tumor tissues were extracted in RIPA lysis buffer (Pierce) supplemented with protease and phosphatase inhibitor cocktail (Cell Signaling). Protein concentration was determined using the BCA assay (Pierce). Lysates were then denatured and samples containing 30 μg proteins were subjected to electrophoresis in 10% SDS-poly-acrylamide gel, followed by transfer to polyvinylidene difluoride membranes (Bio-Rad). After blocking with 5% nonfat dry milk, the membrane was exposed to the primary antibody and incubated at 4 °C overnight. Blots were visualized by labeling with HRP-coupled secondary antibodies (Sigma-Aldrich) and incubation with Amersham ECL western blotting detection reagent (Catalog # RPN2106). Blots were developed on a Konica SRX101A developer and quantified with ImageJ.

**Flow cytometry analysis**. Collected cells were rinsed with PBS containing 0.5% BSA in PBS and the cell pellets were incubated with fluorescence-labeled primary antibodies at 37 °C for 30 min, followed by washing three times with 0.5% BSA/PBS. All antibodies were titrated for optimizing the condition before applying to the experiment. Flow cytometry analysis was performed using the FACS Canto II cytometer (BD) and subsequent analysis was performed using FlowJo (Tree Star, Ashland, OR, USA) software. The gating strategies were based on FSC and SSC properties and were set-up for positive cell populations in FITC, APC channels.

**Human breast cancer tissues**. Fresh frozen HER2 positive and negative breast cancer tissues were provided by the UC Davis Comprehensive Cancer Center Biorepository (IRB # 283665, and IRB # 218204) funded by the UC Davis Comprehensive Cancer Center Support Grant (CCSG) awarded by the National Cancer Institute (NCI P30CA093373) with all patient information blocked except the diagnostic results. Additional human breast cancer pathological samples including the unstained 4-μm-thick sections of formalin-fixed paraffin-embedded (FFPE) paired primary breast cancer and recurrent breast cancers were obtained from the department of pathology, Ohio State University with research IRB approval (#2002H0089).

**Immunohistochemistry (IHC)**. Immunohistochemically staining was done on 4-μm-thick FFPE tissue sections. Briefly, the slides were deparaffinized and rehydrated, and antigens were retrieved for 40 min in a citrate buffer (pH 6.1) at 95 °C. Endogenous peroxidase activity was blocked with 3% $H_2O_2$ solution. The primary antibody incubation was done at 4 °C overnight, followed by the Vectastain ABC Kit (Vector Laboratories) at RT for 30 min according to the manufacturer's instructions. The reaction products were detected using Peroxidase Substrate Kit and counterstained with hematoxylin (Vector Laboratories). Two senior pathologists were in charge of the diagnosis of clinic breast cancers and the evaluation of experimental tumors. The CD47 expression was evaluated by using both staining intensity and percentage of stained cells. The staining intensity was graded as 0: negative; 1: weak; 2: moderate; 3: strong. High expression was defined as strong staining intensity in more than 25% of tumor cells; medium expression was moderate staining intensity in more than 25% of tumor cells; low expression was weak staining intensity in more than 25% of tumor cells or moderate staining in <25% of tumor cells using the ASCO-CAP guidelines[63] was used for the interpretation of HER2 immunohistochemistry (IHC), and HER2 protein expression was scored as negative IHC 0 (no staining or membrane staining that is incomplete and is faint/barely perceptible and within ≤10% of the invasive tumor cells) or negative IHC 1+ (incomplete membrane staining that is faint/barely perceptible and within >10% of the invasive tumor cells); equivocal IHC 2+ (circumferential membrane staining that is incomplete and/or weak/moderate and within >10% of the invasive tumor cells; and or complete and circumferential membrane staining that is intense and within ≤10% of the invasive tumor cells); Positive IHC 3+ (circumferential membrane staining that is complete, intense in more than 10% of the invasive tumor cells). If results were equivocal (2+), reflex testing was performed using in situ hybridization (ISH). In detecting CD47 expression of in vivo irradiated tumors, treated mouse tumors were removed and fixed in 4% paraformaldehyde in PBS for 24 h. The specimens were dehydrated with series ethanol (70%, 95%, and 100%) for 48 h before embedded in the paraffin solution (Polysciences).

**Immunocytochemistry (ICC)**. Cells were seeded on round coverslips and grown to 60–80% confluence, followed by rinsing with PBS, fixing in 4% paraformaldehyde (pH 7.2), and permeabilization with 0.1% Triton X-100 in PBS. The cells were then incubated in a blocking solution for 15 min before incubation with the primary antibody overnight at 4 °C with 1:250 dilutions. Cells were incubated with TR- or FITC-conjugated secondary antibodies diluted 1:1000 in the blocking solution for 1 h at room temperature in the dark and analyzed with confocal microscopy.

**Identification of radiation-associated antigenic proteins**. C57/B6 mouse was used as an immunization host with $5 \times 10^6$ MCF7/C6 cells in 0.1 ml volume injected at the base of tail or footpad per mouse, followed by boosting with the same volume of cells at 14th day. At 5th–7th day following the second challenge the mice were euthanized and the serum was collected to detect antigenic molecules expressed in radioresistant BC cells. In order to distinguish RAAPs from non-specific molecules expressed in normal breast epithelial cells and wild-type MCF7

cells, a pre-clean procedure was performed with the raw serum by the following steps. Two million human normal breast epithelial MCF10A cells were prepared in a solution containing 1 mm EDTA/PBS without trypsin to avoid digesting some of the sensitive proteins, followed by rinsing with PBS twice. The cells were pelleted and then loosen by tapping the bottom of the tube followed by adding 1 ml mouse anti-sera and rotate at 4 °C for 2 h. The mixture was then centrifuged at 9391 × $g$ for 5 min at 4 °C and the supernatants were collected, which was repeated once. The first cleaned antiserum was further cleaned by incubation with wild-type MCF7 cells using the same procedures and repeated six times. The final purified antiserum was then tested by western blot against protein lysate from MCF10A cells, MCF7 cells, MCF7/C6, and RD-BCSC cells that were sorted by breast cancer stem cells markers HER2$^+$/CD44$^+$/ CD24$^-$ as well as aldehyde dehydrogenase (ALDH)[64]. CD47 and HER2 together with other RAAPs in the radioresistant BC cells were identified by western blots or by immunoprecipitation of membrane proteins purified from RD-BCSC cells and the eluted fractions were analyzed via LC/MS. The resulted membrane RAAPs were clustered with a number of proteins and protein functional categories.

**CRISPR editing of HER2 and CD47.** The sgRNAs were designed following the instruction published by Dr. Zhang Lab's CRISPR design software (http://crispr.mit.edu) and the established protocol that has been described in the previous publication[65]. Four Oligos were designed corresponding to the human sgRNAs were synthesized and cloned into lentiCRISPR v2 vector following the Zhang Lab GeCKO pooled library amplification protocol. To minimize the possibility of non-specific targeting, three sgRNA oligos of each targeting gene were synthesized and tested. The sgRNA with the best knockout efficiency determined by western blotting was chosen for subsequent experiments. The sgRNA sequences were used for human and mouse cells as follows:

hHer2gRNA_F: CACCGCGGCACAGACAGTGCGCGTC
hHer2gRNA_R: AAACGACGCGCACTGT CTGTGCCGC
hCD47gRNA_F: CACCGTAAATATAGATCCGGTGGTA
hCD47gRNA_R: AAACTACCAC CGGATCTATATTTAC
mHer2gRNA_ F: CACCGTGATGGCCCTCGCCCCTCGG
mHer2gRNA_ R: AAACCCGAGGGGCGAGGGCCATCAC
mCD47gRNA_F: CACCGCCCTTGCATCGTCCGTAATG
mCD47gRNA_R: AAACCATTACGGACGATGCAAGGGC

The lentiviral particles were generated using 293T cells following the protocol from Addgene. For gene editing, the isolated BCSC sphere cells or MCF7/C6 cells were trypsinized into single cells and plated 1. 2.5 × 10$^5$ cells/0.5 ml per well in the 12-well plates. After incubating for 12 h, 1 ml of virus-containing supernatant with 8 ng polybrene was added to the cells followed by incubation for 6 h. Then, 0.5 ml additional regular medium containing 10% heat-inactivated FBS was added and further cultured for overnight. The infection medium was replaced with 2 ml fresh medium with 10% FBS and cultured for 72 h. Cells were passaged to 60 mm tissue culture dishes and selected by culturing in 0.3 μg/ml Puromysin for 1 week and the knockout of the targeted gene was verified by immunoblotting.

**Luciferase assay.** Cells were transfected with pGL2-basic-CD47 or pGL2-basic-CD47-ΔNF-κB or pGL2-NF-κB luciferase reporters, and luciferase activity was measured by Luminometer (Promega, Madison, WI). For normalization of the reporter transfection efficiency, the total protein concentration of lysates was measured with the BCA Protein Assay kit (Pierce, Rockford, IL) with BSA as the standard.

**Chromatin immunoprecipitation (ChIP) assay.** Cells were cross-linked with formaldehyde (1% final) for 10 min, washed with ice-cold PBS, and collected in SDS lysis buffer (1% SDS, 10 mM EDTA, 50 mM Tris, pH 8. 1). Chromatin was sheared by sonication, pre-cleared with protein G conjugated agarose beads and incubated with anti-p65, anti-c-Rel, or normal IgG at 4 C overnight. After protein G was added for 2 h at 4 °C, the complex was washed in a sequential manner with low and high salt immune complex wash buffers (0. 1% SDS, 1% Triton X-100, 2 mM EDTA, 20 mM Tris-HCl pH 8.1 plus 150 mM NaCl for low salt and 500 mM NaCl for high salt buffers) followed by TE buffer (twice). The DNA-transcription factor interaction was reversed following the addition of NaCl and incubation at 65 °C overnight. After RNase A and Proteinase K digestion, DNA fragments were purified by phenol/chloroform extraction and ethanol precipitation. DNAs were purified and used for PCR with primers specific for the gene promoter region encompassing NF-κB-binding sites. The CD47 primers were:

5′-CGTGGACCAGGACACCTAGG-3′ (sense)
5′-AGGGAAGAGAACCGCATAGG-3′ (antisense)

The PCR amplification of the IκB promoter region (1134/902) was also included as positive. The primers for NF-κB-binding site in the IκB promoter were:

5′-TGTAGCACCCATTAGAAACACTTC-3′ (sense)
5′-TTCTTGTTCACTGACTTCCCAAT-3′ (antisense).

**Preparation of anti-human and anti-mouse CD47 antibodies.** Hybridoma for monoclonal mouse anti-human CD47 antibody (B6H12.2), IgG2b were obtained from ATCC. Monoclonal rat anti-mouse CD47 IgG2a antibody (MIAP301) and hybridoma were obtained from Dr. William Frazier (Washington University

School of Medicine). Both hybridoma cell lines were maintained in IMDM medium with 20% FBS. Antibodies were purified from hybridoma supernatant using protein G Resin from GenScript (Catalog # L00209) following manufacture standard procedures. Samples were then further concentrated 10–20-fold using Microsep centrifugal devices from Pall Life Sciences. Concentrations of purified IgG were determined by measuring the absorbance at OD$_{280}$.

**Phagocytosis assay.** Both Human monocyte THP1 cells and Mouse Mφ RAW 264.7 cells (ATCC, TIB-71) were maintained in a 5% CO2 incubator at 37 °C[66] at an approximate density of 1 × 10$^6$/ml. About 0.8 × 10$^6$ cells/ml of RAW 264.5 cells or 1 × 10$^6$ THP1 cells were seeded onto a 6-well plate. After 24 h of incubation RAW 264.7 cells were activated by treating with 0.1 μg/ml of lipopolysaccharide (LPS) for 24 h. Monocyte THP1 cells were differentiation by Phorbol 12-myristate 13-acetate (PMA) (40 nM) for 48 h. Activated macrophages were stained with Dio at final concertation 40 nM in a medium for 20 min followed by rinsing three times with medium. Cancer cells were labeled with 1 μM DDAO (5 mM stock solution in DMSO stored at −20 °C) in PBS at 37 °C for 15 min and washed with PBS containing 1% FBS. DDAO-labeled target cells (1 × 10$^6$) were added to DIO-stained Mφs and incubated in a final volume of 2 ml at 37 °C for 2 h. Following the incubation, Mφs and target cells were harvested with three times EDTA-PBS wash followed by 0.25% trypsinization. Phagocytosis was assessed by evaluating the dual-labeled cells (DIO$^+$/DDAO$^+$), which represents phagocytized breast cancer cells by mature Mφs, via flow cytometry. The FlowJo software was used for analysis.

**Clonogenic survival assay.** Clonogenic survival assay was performed following radiation with or without treatment (Lapatinib, 10 μM for 72 h; anti-CD47 antibody 10 μg/ml overnight). The treated cells were cultured for 10–14 days and colonies were fixed, stained with Coomassie blue stain. Colonies containing more than 50 cells were counted as surviving clones and normalized to the plating efficiency of each sham-treated tumor cell line.

**Gap-filling rate assay.** Gap-filling capacity was measured with 1 × 10$^6$ cells grown in each of 6-well plates to 100% confluence followed by 24 h cell starvation. Then the gap was created by scraping the dish diagonally with a sterile pipette tip. The cells were either left untreated or treated with 10 μg/ml of IgG or anti-CD47 antibody during the duration of the experiment. The filling capacity was monitored for 72 h and representative images were taken at day 0 (day of scraping) and day 3.

**Tumorsphere formation.** Cells were sieved with 40 μm cell strainers (Catalog # 352340. Corning) and single-cell suspensions were seeded into low-attachment 60-mm Petri dishes at a density of 1000 cells/ml. The cells were grown in serum-free mammary epithelial basal medium (MEBM), supplemented with B27 (Catalog # 17504-044. Life Technology), 20 ng/ml EGF (Catalog # 4022-500. Bio-vision), 20 ng/ml basic-FGF (Catalog # 13256-029. Invitrogen), and 4 μg/ml heparin (Catalog # 80603-686 EMD MILLIPORE). Cells were cultured for 10 days and tumorspheres were counted under light microscopy.

**Transwell invasion assay.** Aliquots (0. 4 ml) of Matrigel (Catalog # 356231. BD Biosciences) were diluted in coating buffer: 0.01 M Tris (pH 8.0), 0.7% NaCl to the final concentration of 200–300 μg/ml. About 100 μl were loaded into the upper chamber of 24-well transwell (Catalog # 3422. Costar) and incubated at 37 °C for gelling about 1 h. Carefully remove the remaining coating buffer from the permeable support membrane without disturbing the layer and then add cells (2.5 × 10$^4$/ml). The lower chamber was filled with 800 μl of MEM media containing 5 μg/ml fibronectin (Catalog # SC-29011. Santa Cruz). The transwell with differently treated cells was incubated for 48 h and stained with Diff-Quick Stain kit (Catalog # K7128. IMEB INC).

**Preparation of F (ab′)$_2$ fragments.** CD47 F (ab′)$_2$ fragments were produced by Papain resin cleavage of IgG using a F (ab′)$_2$ preparation kit (Catalog # 786272. G-Bioscience) according to the manufacturer's recommendations and reported procedure[67]. Following papain digestion, the Fab fragment was separated from undigested IgG and the Fc region with the Protein A Spin Column from the Fab Fragmentation Kit The SpinOUT GT-600 desalting columns were supplied to ensure the initial antibody sample to be the optimal condition for Fab fragmentation and the purified anti-mouse CD47 IgG was collected for the in vivo mouse tumor treatment.

**Radiation of BC cells with CRISPR-KO CD47 and/or HER2.** For in vivo test of tumor inhibition efficacy by radiation with deficient CD47 and HER2 status, 5 × 10$^5$ 4T1/C2 cells with CRISPR-knockout of CD47 (CD47$^{-/-}$), HER2 (HER2$^{-/-}$) or double (CD47$^{-/-}$/HER2$^{-/-}$) were implanted into the mammary fat pads of BALB/c ($n = 6$ per group). Tumor growth was assessed by measuring the tumor volume every 2 days starting day 7 till the tumor volumes reached the limitation. To assess the radiosensitivity of tumors with different status of CD47 and HER2, local tumor radiation was delivered with 5 Gy to each tumor at day 9 and tumor growth was measured every other day till control tumors reached the maximal limitation.

**RT of mouse BC with anti-CD47 and/or HER2 treatment**. For in vivo tests of tumor inhibition with a single CD47 antibody in the presence or absence of radiotherapy, anti-CD47 treatment followed the protocol by Willingham et al[20] with some modifications. Eight weeks old immune-competent (BALB/c) female mice (Charles River Laboratories, Sacramento, USA) were injected with $1 \times 10^5$ mouse breast 4T1 cells into the 4th mammary glands. Fifty microliters PBS containing 100 µg of control IgG or anti-CD47 F (ab′) fragments were injected into the tumor site (day 1) or into the tumor tissues (day 15) and repeated every other day until the end of experiments. Tumor volumes were monitored every 5 days. For radiation treatment, anti-CD47, or radiation combined with anti-CD47, the animals bearing 4T1 tumors were randomly divided into various treatment groups and a control group (5–10 mice per group). Tumor local radiation started when tumor volume reaches 200 mm³ with FIR (5 Gy/day/tumor for four fractions using micro-beam IR source; total dose = 20 Gy) combined with three times tumor injections of anti-CD47 F (ab′). The radiosensitivity of the in vivo irradiated tumors with the presence or absence of anti-CD47 F (ab′) was evaluated by measuring tumor volume at the end of experiments. The animals were euthanized when the tumors were ~1400 mm³ or when the animals seemed to be uncomfortable even when the tumor was smaller than 1400 mm³ to comply with UCD IACUC regulations for use of vertebrate animal in research. The animal use and care protocol of in vivo radiotherapy was approved by the Institutional Animal Use and Care Committee of the University of California Davis (IACUC 15315).

For in vivo tests of tumor inhibition with dual antibodies to CD47 and HER2 in the presence or absence of radiotherapy, eight weeks old immune-competent (BALB/c) female mice (Charles River Laboratories, Sacramento, USA) were injected with $1 \times 10^5$ mouse breast 4T1 cells into the 4th mammary glands. When tumor volumes reached about 200 mm³, mice were randomly divided into four groups (6 mice per group). PBS, IgG, anti-CD47 F (ab′) fragments (100 µg), or Herceptin (5 mg/kg) were injected into tumor tissues 4 h before local radiotherapy (5 Gy per day for 2 days). Injections were performed and tumor sizes were monitored every other day till the end of experiments. Mice were euthanized when the tumors were ~1400 mm³ or when the animals seemed to be uncomfortable even when the tumor was smaller than 1400 mm³ to comply with UCD IACUC regulations for use of vertebrate animal in research. The animal use and care protocol of in vivo radiotherapy was approved by the Institutional Animal Use and Care Committee of the University of California Davis (IACUC 15315).

**Tumor-infiltrated macrophages and macrophage phagocytosis**. GFP-expressing mouse breast cancer 4T1 cells were implanted into 4th mammary fat pads of BALB/c mice and treatment was started when tumors achieved about 200 mm³ by tumor injection of 50 µl PBS containing 100 µg of control IgG, anti-CD47 F (ab′) fragments, Herceptin or both of antibodies every other day for three times. Tumor local RT was delivered on days 10 and 11 with 5 Gy/day/tumor for two fractions, total dose = 10 Gy, and experiments were ended 2 days after the termination of antibody treatments. Tumors were extracted and FFPE sections were prepared for immunofluorescence staining to follow the standard procedure: the slides were deparaffinized and rehydrated, and antigens were retrieved for 40 min in a citrate buffer (pH 6.1) at 95 °C. To remove autofluorescence, slides were socked in de-autofluorescence solution (0.5 M CuSO₄, 0.05 M NH₄COOH in H₂O) at RT for 48 h, rinsed with water 5 min 3times, then blocked with 1:100 horse serum. The primary antibody incubation was done at 4 °C overnight: anti-GFP (1:100; Dako), anti-CD11b (1:100; Millipore); After washing, slides were incubated with 1:250 diluted fluorescent conjugated secondary antibodies (Rhodamine for CD11b, Alexa Fluor 488 for GFP) at RT for 1 h and then mounted with Vectashield Antifade solution containing DAPI (Vector Laboratories, Burlingame, CA, USA). Macrophage-mediated phagocytosis was detected with fluorescent microscopy using Axiovision software (Zeiss, Germany) and microphotographs were quantified by ImageJ.

**Statistical analysis**. All of the experimental data obtained from in vitro cellular studies and animal tests are presented as mean ± SE, analyzed using the two-tailed Student t-test for two groups or ANOVA for multiple groups. The statistical significance of Kaplan–Meier survival curves was assessed with a Mann–Whitney test. The number of independent experiments and the replicates were indicated in the figure legends. P-values < 0.05 were considered to be significant and are indicated by asterisks as follows: $*P < 0.05$, $**P < 0.01$, $***P < 0.001$, $****P < 0.0001$.

**Reporting summary**. Further information on research design is available in the Nature Research Reporting Summary linked to this article.

## Data availability

The data that support the findings of this study are within the Article, Supplementary Information, or available from the corresponding author upon reasonable request. Source data are provided with this paper.

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

## Acknowledgements

We thank Dr. Irmgard Feldman for sharing human breast cancer specimens. We acknowledge Dr. William Frazier for kindly providing anti-mouse CD47 MIAP301 hybridoma, Dr. Qian Chen for the assistance of in vivo macrophage phagocytosis, and Mr. Shuaib Juma and Dr. Julian Perks for conduction of radiation with in vivo mouse breast tumor model. This work was supported by NIH RO1 CA213830 (J.J.L.) and CA224900 (H.-W.C.), University of California Davis Cancer Center Cancer Immunology Pilot Grant Support, and University of California Davis NCI-designated Comprehensive Cancer Center supported by the CCSG Grant awarded by the National Cancer Institute (NIH NCI P30CA093373).

## Author contributions

D.C-G, B.X., M.F., L.S., A.M.M., W.J.M., H.-W.C., K.S.L, R.R.W., and J.J.L. designed the concept, developed methodology, acquired, analyzed and interpreted data, and provided administrative, technical, or material support. L.Z., D.J.G., C.-x.P., S.X., Y.Z., R.S., T.L., L.L., and J.H. characterized the radioresistant breast cancer cells and in vivo irradiated tumors. B.X., C.M., D.J.G., T.G., W.Z., N.J., T.-y.L., O.M.O., and M.F. performed animal tests on radiation combined antibody blockade of CD47 and HER2. B.X., M.F., and H.C. established CRISPR KO cell lines. A.T.V., G.W., M.F., and D.J.G. supervised the in vivo radiation settings. C.S. and S.A. guided the study of clinical samples of breast cancer patients. Y.Z. and R.S. contributed to the pathological studies on the clinical samples. J.M.W., W.J.M., and A.M. guided the studies on tumor tests of antibody dual inhibition on CD47 and HER2 and phagocytic analysis. D.C-G., B.X., A.T.V., S.X., L.Q.S., H.-W.C., K.S.L., R.R.W., and J.L.L. reviewed the data, wrote the manuscript, and supervised the study.

## Competing interests

The authors declare no competing interests
