## [Peer Review File · Nature Communications]

Reviewers' Comments:

Reviewer #1 (Remarks to the Author)

Authors demonstrate that CD47 expression is regulated by NFkappaB-HER2 axis in response to radiation. Some of the data is novel in relation to radiation-induced signal transduction pathways. Methodology is appropriate, however, many of the Western blot data and immunohistochemistry photomicrographs are not convincing and questionable. There are several major concerns:

- The title of the paper refers to immune privilege and study concludes that blocking CD47 and HER2 can enhance tumor response to radio-immunotherapy, however, no data is reported here to demonstrate that there is immune privilege environment in radiation treatment setting and in what level the radio-immunotherapy can be effective.

- The promoter reporter studies of CD47 is not convincing. It lacks positive driver control such as p65 mediated transcriptional up regulation of CD47 (fig 1d and 1e). Chip assay data are not convincing, it is important to demonstrate by mutating the NFkappaB binding, EMSA should be added.

- Data such as Figure 4c, and 4 a re not convincing.

- The pathway is premature as it is not clear how radiation activates NFkappaB? Is TNF involved for NFkappaB? How does Her2 expression unregulated in response to IR?

- It is well-known that tumors have increased expression of CD47 in order to strengthen the inhibitory signals through SIRPa and to more potently inhibit phagocytosis mediated by Fcγ receptors. Figure 5 demonstrates an increase in macrophage phagocytosis, however, it is not supported by molecular interaction data such as phosphorylations kinetics of SIRPa.

- Tumor response data in figure 6 lacks to demonstrate the role of immune modulation and hence it is not conclusive to report that abrogating the function of CD47 and HER2 can enhance radio-immunotherapy.

Minor concerns

- What is SSC-A in Figure 1b? There is no time course in figure 1b as indicated in the legend.
- Authors claim there is induction of CD47 in tumors, but based on S1a, it does not.

Reviewer #2 (Remarks to the Author)

The article by D. Candas et al links CD47 and HER2 expression with adaptive immune-tolerance leading to radioresistance of many cancers. The proposed mechanism is a complex interplay between irradiation and a NF-κB-HER2-CD47 loop. The work has potential implications for human cancer treatment given the HER2 directed therapies are commonly available, irradiation is a mainstay of cancer treatment, and anti-CD47 monoclonal antibodies are currently being in early phase of clinical development.

The following points should be addressed by the Authors:

1. The citation of ref 21 (Cao N et al., page 4 of the Introduction and page 11 in the Discussion) is not completely accurate, since the cited work states that that NF-kappa B is required for radiation-induced HER2 transactivation in cells that were poor expressers of HER2.
2. The sentence on page 4 after line 7 is a summary of the results to be presented in the article. It should not appear in the Introduction
3. On page 5, after showing the data presented in the 6 panels of Fig 1, the Authors conclude that "The results demonstrate that induction of CD-47 plays a role in adaptive tumor radioresistance". However, what they actually showed is that CD47 expression was enhanced by exposure to

irradiation of MCF7 and MCF7/C6 and other cell lines (HCT116, HepG2 and U251). The different expression of CD47 in primary tumors and metastatic lesions is not per se enough to conclude that CD47 plays a role in adaptive tumor resistance. In addition, how many of the 36 samples were primary tumors and how many were metastatic lesions? Were they matched primary and relapse samples from the same patients? In brief, what the Authors show in the Figure and in the text is that irradiation triggers expression of CD47 in cell lines and in tumors in mice, and that CD47 expression is more frequent in metastatic lesions in women (with the caveat that 36 cases are a very limited sample to draw any firm conclusion).

4. In Figure 3a the expression of HER2 by MCF7 is low (in some MCF7 cell lines is actually nil), while CD47 expression looks high and even higher than in the HER2-amplified BT474. In addition, in Fig 3b the highest CD47 expression is observed in tumor T1, which has minor HER2 expression, and is absent in T2 that has the highest level of HER2 expression. The Authors should clarify whether the "HER2+ tumors" are actually amplified ones, and whether the samples were obtained from primary or metastatic lesions. The latter is particularly relevant in light of the data presented by the Authors in Fig 1 about the different expression of CD47 in primary and metastatic lesions.

5. It is unclear to this Reviewer what the sentence on page 6 means ("Eight out of 13 HER2 positive tumors were CD47 high, 2 were CD47 low, and 3 showed a medium level; whereas, 10 of 13 HER2 negative tumors were low in CD47 expression, 3 showed medium, and none showed high CD47 expression (Fig. 3c, right panel)."). Where these tumors are coming from? Are they part of the batch of 36 that are cited in Fig. 1? Again, what was their origin (primary lesion v. metastatic). Also, there is no logical link between the data illustrated in Fig. 3a-c, those presented in Fig. 1 about the role of irradiation and the "possibility that CD47 is enhanced in recurrent and metastatic breast cancer" as stated in lines 10-11 of page 6. CD47 is induced by IR and is found in HER2 overexpressing tumors (are they amplified?).

The data shown on Fig 3d-e are not particularly convincing, and it is unclear why the Authors selected the 231/C5 cells instead of using again the MCF7/C6 as in Figure 1. Finally, what is the link between the experiment with irradiated 4T1 tumors (Fig 3f) and the co-expression of CD47 and HER2 illustrated in the recurrent and primary tumors of Fig. 3g?

6. Given the relevance of the role of HER2 in the entire work, it is strongly suggested that IHC were adopting the criteria for HER2 scoring described in ASCO-CAP guidelines in 2013.

7. The sentence in the last line of page 6 starting with "Notably, both basal and IR-induced..." is repeated in the immediately following sentence. Please condense and rectify.

8. This reviewer can hardly see an effect of Lapatinib or Herceptin on C6 in Figure 4b. In Fig. 4c there is no effect of Herceptin on CD47 expression in BCSC exposed or un-exposed to irradiation. Also, there is a discrepancy between the CD47 basal expression in SKBR3 in Fig. 3a and that in Fig 4d, where it is elevated only after irradiation. This is not irrelevant given that the experiment is key to show the link between HER2 amplification and CD47. The experiment in Fig. 4h and 4g again goes back to the radiation resistant MCF7/C6, but co-activation of HER2 and CD47 in IR-resistant cells was presented only for 231/C5 in Fig. 3d and 3e. Why this lack of consistency in the selection of the model cell line?

9. On page 7 the Authors state "The present data further revealed that the NF- κ B-HER2-CD47 pathway can be induced by IR via NF- κ B activation, indicating the complexity of acquired tumor resistance." However, this is not what the data in Fig. 4 show. There is no evidence as to the role of NF- κ B in the experiments illustrated in Fig. 4 and discussed in page 6 and 7 of the text. Overall the scheme sketched in Fig. 4i is tempting, but not demonstrated. Also, is it to be applied to bona fide HER2+ tumors that are almost always c-erbB2 amplified (SKBR3), or to IR resistant cell lines that bear co-activation of HER2 and CD47? Or to any cancer cell exposed to irradiation? The models presented in the different experiments correspond to completely different clinical conditions in man, and open completely different therapeutic scenarios.

10. To this Reviewer Fig. 5e-g show that irradiation of MCF7 has no effect on phagocytosis, and that anti-CD47 actually lowers the phagocytosis itself. It is unfortunate that there are no data on MCF7 co-treated with anti-CD47 and Herceptin. A similar pattern on phagocytosis is observed with MCF7/C6 and anti-CD47 (Fig. 5f), or anti-CD47 and Herceptin (Fig. 5g). How can this be used to conclude that irradiation "enhanced CD47 expression can compromise macrophage phagocytosis" (lines 7-8 of page 8) is frankly unclear to the reviewer. Do the Authors want to suggest that

irradiation increased CD47 to such a level that the antibodies are unable to block the down modulation of phagocytosis? This is not what could be expected from the evidence presented in the previous sections. Up to this point and to the evidence of Fig 5 the hypothesis was that CD47 - the "don't eat me signal"- was up-regulated with HER2 in irradiated cases, leading to immune evasion that would consist of avoidance of phagocytosis. Maybe the Authors should check that the conduct of the experiments caused an irradiation-dependent toxicity on macrophages. That would explain the findings and would call for a different experimental approach.

11. With a somewhat logic leap forward, the Authors show in Fig 6 that indeed CD47 expression promoted the aggressive phenotype of MCF7/C6 that could be inhibited by anti-CD47. Importantly, the combination of irradiation, anti-CD47 and Herceptin, that did nothing on phagocytosis, eliminated clonogenic resistant cells. No idea of the mechanism of action for such an effect. Have the Authors any hypothesis to share?

12. Experiments illustrated in Fig. 6f-h require some clarification. How large were the tumors at time 0 for the group exposed to anti-CD47 on D1? At week 1 they already were significantly smaller than those in the control group and in the anti-CD47 D15 (Fig. 6f). The interpretation of the results of Fig. 6h is based on a crude difference of 20% survival in a single experiment involving 10 animals per group. The 2 more animals surviving is a very small difference for the conclusion about the synergy of low dose radiation and anti-CD47 therapy. Finally, it is unfortunate that there is no animal data for the triple combination of HER2 directed therapy, anti-CD47 and irradiation.

13. The Discussion has a number of statements that are not always supported by the data. In addition there are entire paragraphs (last of page 11 and first of page 12) that recapitulate the Results with direct reference to the figures in an unneeded rehearsal of data that were already extensively described in the Results. Here is a brief list of comments for the Authors:

- a. As many other cancer treatment, irradiation causes cell death (including immunogenic cell death) and triggers mechanisms of resistance. The Authors show that CD47 expression may be one such mechanism that masks tumor cells from the phagocytic activity of macrophages. The hypothesis is sound, but the limiting factor of their observation is that combination of irradiation and therapies directed to CD47 and HER2 (Fig. 5) is far from being so active (already discussed in my point 10 above). The first paragraph of page 10 should take into account this aspect. Also, it is unclear what evidence in the present article supports the concept stated in lines 16-17 that "...a dual inhibition of CD47 and HER2 may enhance the abscopal effect of radiotherapy". The dual inhibition already works very little locally in the presented experiments. Based on what do the Authors expect an increased abscopal effect at a distance?
- b. The sentence in page 10 line 21 "Recently cetuximab..." appears incomplete and out of context.
- c. The parallel between the co- regulation of HIF1a and CD47 on page 11 has nothing to do with present work
- d. The last sentence in the Discussion ("...local tumor radiotherapy with immune blockade of CD47 and HER2 may be an effective approach to enhance the tumor response to radiotherapy") suggests that every irradiation to any type of cancer irrespective of the basal HER2 expression should be combined with anti-CD47 and anti-HER2 therapy. Frankly, this is not supported by the data.

Point to point response

Reviewer #1 (Remarks to the Author):

Authors demonstrate that CD47 expression is regulated by NF- κ B-HER2 axis in response to radiation. Some of the data is novel in relation to radiation-induced signal transduction pathways. The methodology is appropriate; however, many of the Western blot data and immunohistochemistry photomicrographs are not convincing and questionable. There are several major concerns:

Response: We thank you for your evaluation on the novelty of this work. We very appreciate your pointing out the weaknesses that are critical for improving the overall quality. In the revised manuscript many experiments have been repeated and substantial data have been added. The title has been changed to precisely present the findings.

1. The title of the paper refers to immune privilege and study concludes that blocking CD47 and HER2 can enhance tumor response to radio-immunotherapy; however, no data is reported here to demonstrate that there is immune privilege environment in a radiation treatment setting and in what level the radio-immunotherapy can be effective.

Response: We totally agree with your critiques. Following your suggestion, a substantial amount of data has been added. In Fig.4A, we show that MCF7/C6 cells demonstrated an enhanced anti-phagocytosis ability with increased expression of CD47, which can be partly rescued by anti-CD47 treatment or CRISPR-CD47 (Fig. 4i). To further test the potential synergy by dual inhibition of CD47 and HER2 with radiation, we have successfully established CRISPR-edited dual deficiency of CD47 or HER2 compared to CRISPR-KO of either receptor and tested in vivo mouse syngeneic breast tumors showing an enhanced synergy in tumor inhibition by radiation combined with dual gene KO than blocking either receptor alone. These results are now presented in Fig. 6.

2. The promoter reporter studies of CD47 is not convincing. It lacks positive driver control such as p65 mediated transcriptional up regulation of CD47 (1d and 1e). ChIP assay data are not convincing, it is important to demonstrate by mutating the NF- κ B binding, EMSA should be added.

Response: We very appreciate your insightful comments. The luciferase reporter assay and ChIP assay previously shown in Figs. 2d, e, and 2f are now moved to Figs.3g, 3i, and 3l in the revised manuscript. With your suggestion, we again repeated the experiments of luciferase reporter assays to confirm the results. In the ChIP assay, we added groups of total chromatin as a positive control, anti-p50 was also included serving as one more evidence to identify that radiation promoted the recruitment of NF- κ B to CD47 Promoter. Clearly, as shown in Fig. 3g, TNF α significantly increased the luc activity in cells with wildtype CD47, whereas the increment was impeded by a mutation of NF- κ B binding motif in the CD47 promoter region. In Fig.3h, radiation-promoted CD47 expression was reduced by NF- κ B inhibition (IMD) or mutation of NF- κ B binding motif in the CD47 promoter region. In addition, IMD was further introduced to block NF- κ B signaling, immunofluorescence and western blot were subsequently used to detect the CD47 expression at protein levels.

EMSA is used traditionally for analyzing protein-DNA interactions in vitro making it hard to quantitate the transcriptional activity of a specific gene promoter. It needs to perform the super shift assay with antibody to be certain of protein identity in a complex. ChIP method are currently used by many studies and used to monitor transcriptional regulation through specific binding motifs. The ChIP assay method allows analysis of DNA-protein interactions in living cells by treating the cells with formaldehyde or other crosslinking reagents in order to stabilize the interactions for

downstream purification and detection. In addition, ChIP can be carried out separately on chromatin that was immunoprecipitated with antibodies to proteins for determining protein-DNA interactions [1-5]. Thus, EMSA-identified protein-DNA binding is usually to be confirmed using ChIP assay [6]. The new data presented in Fig. 3g-k support that that NF-κB binds to the CD47 promoter region and promotes CD47 expression radiation response.

3. Data such as Figs. 4c, and 4 are not convincing (Fig.4C is not very convincing since it shows that radiation induced CD47 expression which is absence in Herceptin treated cells but why radiation did not induce CD47 expression in the HER2- BCSCs?)

Response: We have conducted an array of experiments suggested by both reviewers on this point and re-arranged the figures. Fig. 4c was moved and shown as Fig. 3c in the current version. Data shown in Fig.2 demonstrated the co-expression of CD47 and HER2 in diverse breast cancer cells and clinical samples. Importantly, CD47⁺ population were increased by radiation in HER2-expressing MCF7/C6 cells and SKBR3 cells, whereas the radiation-induced increment of CD47⁺ population was significantly impeded in the presence of lapatinib, an inhibitor that targets the TK domain of HER2 for blocking HER2-mediated signal transduction. This finding was further supported by lack of radiation-induced CD47 expression in Herceptin-treated HER2⁺ breast cancer stem cells (HER2⁺ BCSCs) and in HER2⁻ BCSCs (Fig. 3c).

4. The pathway is premature as it is not clear how radiation activates NFκappB? Is TNF involved for NFKappaB? How does Her2 expression upregulated in response to IR?

Response: Thank you for your comments. Question (a). In addition to immune responses, NF-κB is well demonstrated in genotoxic stress conditions including ionizing radiation which is mediated by radiation-associated redox imbalance and protein tyrosine kinase activation. We and other researchers have reported that [7,8] radiation-generated ROS activates NF-κB mainly due to two ways: cell receptors activation (such as HER2 and EGFR) due to ROS mediated kinases activation and DNA damage induced IKK regulation [7-9], both induce the traditional NF-κB signaling-mediated expression of the downstream effector genes including CD47 in this study. Since it is known that HER2 regulates NF-κB signaling via PI3K/AKT, radiation induced NF-κB is predominantly enhanced in HER2 expressing cancer cells [10, 11]. Here we further revealed that radiation induced CD47, an immune evasion-associated gene containing NF-κB binding motif in the promoter region, is also dominantly enhanced in HER2-expressing tumor cells, new evidence indicating that cancer cells can simultaneously activate two receptors with different functions via NF-κB regulation. This proposed signaling network is now illustrated in Fig. 7 that is attached here for your convenience.

Question (b). Yes, TNF-α is well defined in NF-κB activation. A report recently published in Nature [12] indicates that CD47 expression can be induced by the pro-atherosclerotic factor TNF-α which supports our finding that NF-κB is the key transcription factor controlling CD47 expression since TNF-α is a well-defined upstream factor for

inducing canonical NF- κ B transactivation [13-16]. TNF- α is shown to function through two distinct cell surface receptors, TNFR1 and TNFR2 and the induced NF- κ B activity involves the five mammalian NF- κ B/Rel proteins: c-Rel, NF- κ B1 (p50/p105), NF- κ B2 (p52/p100), RelA/p65, RelB. In the absence of TNF- α stimulation, NF- κ B is combined with the inhibitor I κ B in the cytoplasm. TNF-induced activation of NF- κ B largely relies on phosphorylation dependent ubiquitination and degradation of inhibitor of kappa B (I κ B) proteins. The inhibitor of κ B kinase (IKK) complex, a multiprotein kinase complex is responsible for the TNF- α induced phosphorylation of I κ B. Although this pathway is involved in the regulation of a wide spectrum of biological processes including cell proliferation and differentiation, our findings demonstrate the first evidence that HER2-promoted cell proliferation can be coordinated with CD47-mediated immunotolerance in radioresistant breast cancer cells. Question (c). HER2 promoter contains NF- κ B binding motifs and radiation induced HER2 expression has been reported by our group [17, 18]. As shown in Figs. 2a-e in the current manuscript, both of HER2 expression and HER2⁺ population were remarkably enhanced in the resistant cancer cells compared to the counterpart controls. In addition, HER2 expression in the recurrent breast tumor tissues was obviously elevated compared to the primary tumors. These results support that HER2 expression is inducible by radiation via NF- κ B regulation which also affects cell immunotolerance via CD47 expression.

5. It is well-known that tumors have increased expression of CD47 in order to strengthen the inhibitory signals through SIRP α and to more potently inhibit phagocytosis mediated by Fc γ receptors. Figure 5 demonstrates an increase in macrophage phagocytosis, however, it is not supported by molecular interaction data such as phosphorylation kinetics of SIRP- α .

Response: We very appreciate your insightful comment. CD47 was first identified as a tumor antigen on human ovarian cancer in the 1980s [19]. Since then, CD47 has been found to be expressed on multiple human tumor types and overexpression of CD47 was reported to enable tumors to escape innate immune system surveillance through evasion of phagocytosis. By binding and activating signal regulatory protein- α (SIRP α), an inhibitory protein expressed on the surface of myeloid cells, CD47 serves as an anti-phagocytic or “don’t eat me” signal. Liu et al found that both of anti-CD47 and anti-SIRP α could suppress tumor growth in mice with intact immune system [20]. In other studies, anti-CD47-mediated phagocytosis enhancement was confirmed by using diverse models treated with anti-CD47 antibody and detecting phagocytosis index [21-24]. Consistently, we employed different strategies including CRISPR/cas9-based CD47 knockout or anti-CD47 antibody-mediated CD47 blockade to determine the effect of CD47 expression on the immune evasion of resistant breast cancer cells (4). Diverse in vivo models were also introduced to confirm the efficacy of blocking CD47 in treating resistant breast tumors (5 and 6). Notably, we not only determined the anti-tumor effects of CD47 depletion in this study, but also revealed the HER2-NF- κ B-regulated CD47 expression, suggesting that resistant breast cancer cells may escape radiation treatment and immune surveillance due to HER2-induced intrinsic resistance and CD47-mediated immune evasion, dual inhibition of CD47 and HER2 is a potential effective strategy in breast cancer radiotherapy. It is very hard to quantitate SIRP-phosphorylation with in vivo tumors and the in vitro data using in vitro activated macrophages could not represent the in situ activation in the tumor microenvironment. Therefore, CD47 mediated anti-phagocytosis via SIRP- α phosphorylation on macrophages and interaction between SIRP- α and SHP-1 has been well demonstrated. Our new data (Figs. 4-6 together with Figs. S6-S10) are supportive to the conclusion that macrophage-mediated phagocytosis was correspondingly responded to altered CD47 expression levels (references have been added in the revised manuscript).

6. Tumor response data in figure 6 lacks to demonstrate the role of immune modulation and hence it is not conclusive to report that abrogating the function of CD47 and HER2 can enhance radio-immunotherapy.

Response: We agree with your comments. A substantial amount of in vivo tests has been conducted to verify the synergetic tumor inhibition by anti-HER2 and anti-CD47 treatment. Since pro- (immune suppression) and anti-tumor (abscopal effect) are both evidenced in the literature, we attempted to identify the pro-tumor factors to which new modality may be invented to enhance the radiation/immunotherapy efficacy that has been evidently supported by an array of studies ^[25-27]. The new data shown in Figs. 5 and 6 in the revised work indicate that dual gene deficiency by CRISPR-editing (Fig. 5) or by dual antibody blocking (Fig. 6) enhanced the synergy with radiation compared to radiation with a single receptor inhibition. It should be noticed that radiation combined with either receptor depletion also generated a significant inhibition compared to radiation alone (Fig. 5c). Interestingly, dual antibody treatment plus radiation almost totally eliminated tumor growth in the mouse orthotopic breast cancer, which also agreed with an enhanced scale of macrophage-mediated phagocytosis (Fig. 6).

Minor concerns

1. What is SSC-A in Figure 1b? There is no time course in figure 1b as indicated in the legend.

Response: Thank you for your comment. (a) The SSC-A is a gating strategy for flow cytometry. SSC (side scatter) parameter is a measurement of the amount of the laser beam that bounces off of particulates inside of the cell. SSC-A is helpful for identification of cells with varying complexity.(b) The legend of Fig. 1b has been corrected.

2. Authors claim there is an induction of CD47 in tumors, but based on Figure S1a, it does not.

Response: Thank you for your carefully evaluation. We previously did not use Fig. S1a to show radiation induced CD47 expression. The data in the original Fig. S1a were shown to confirm CD47 expression in breast tumor versus surrounding normal tissue in breast cancer patients. Consistent with recent clinical studies ^[28-30], our results demonstrated the elevated expression of CD47 protein in 4 from 5 freshly-dissociated clinical breast tumors compared with the corresponding non-tumor breast tissues from the same patients. In the current manuscript, we replaced Fig. S1a with Fig. 1e to show the difference of CD47 expression between HER2⁺ (IHC positive, FISH positive) and HER2⁻ (IHC negative, FISH negative) breast tumors. As shown in Fig.1e, no CD47 protein was detected in HER2⁻ samples while 3 out of 4 HER2⁺ tissues showed CD47 expression. Although the expression of HER2 and CD47 was not shown in a positive correlation, unknown mechanisms may exist in HER2⁺ cells which affects CD47 protein levels and needs to be further studied in the future. Together with the in vitro comparison of CD47 expression in multiple HER2⁺ or HER2⁻ cell lines (Fig. 1d), we suggest that the induction of CD47 is linked with HER2 status.

Reviewer #2 (Remarks to the Author):

The article by D. Candas et al links CD47 and HER2 expression with adaptive immune-tolerance leading to radioresistance of many cancers. The proposed mechanism is a complex interplay between irradiation and a NF-κB-HER2-CD47 loop. The work has potential implications for human cancer treatment given the HER2 directed therapies are commonly available, irradiation is a mainstay of cancer treatment, and anti-CD47 monoclonal antibodies are currently being in early phase of clinical development.

Response: We very appreciate your careful evaluation and insightful comments to this study.

The following points should be addressed by the Authors:

1. The citation of ref 21 (Cao N et al., page 4 of the Introduction and page 11 in the Discussion) is not completely accurate since the cited work states that that NF-kappa B is required for radiation-induced HER2 transactivation in cells that were poor expressers of HER2.

Response: Thanks again for pointing out. We have thoroughly revised the Introduction. Related papers on HER2-NF- κ B pathway via the PI3K/AKT signaling have been cited in the revised manuscript^[17, 31, 32].

2. The sentence on page 4 after line 7 is a summary of the results to be presented in the article. It should not appear in the Introduction.

Response: Thank you for your insightful comments. We have rewritten this part of result summary in the Introduction.

3. On page 5, after showing the data presented in the 6 panels of Fig 1, the Authors conclude that “The results demonstrate that induction of CD-47 plays a role in adaptive tumor radioresistance”. However, what they actually showed is that CD47 expression was enhanced by exposure to irradiation of MCF7 and MCF7/C6 and other cell lines (HCT116, HepG2 and U251). The different expression of CD47 in primary tumors and metastatic lesions is not per se enough to conclude that CD47 plays a role in adaptive tumor resistance. In addition, how many of the 36 samples were primary tumors and how many were metastatic lesions? Were they matched primary and relapse samples from the same patients? In brief, what the Authors show in the Figure and in the text is that irradiation triggers expression of CD47 in cells lines and in tumors in mice, and that CD47 expression is more frequent in metastatic lesions in women (with the caveat that 36 cases are a very limited sample to draw any firm conclusion).

Response: Thank you for your comments. We have rewritten the Results including section 1. The clinical data have been expanded and further reviewed and graded according to WHO standardization which has been conducted by Dr. Yanhong Zhang, a pathologist at UC Davis Cancer Center. She is credited as a co-author for the revised manuscript. It turned out to be difficulty to obtain paired primary and recurrent tumor sample because radiotherapy is recommended mostly in adjuvant settings according to NCCN breast cancer guideline and few patients were willing to take a second biopsy when tumor recurrence, especially with distant metastasis. Samples in Fig. 1f were obtained from pathological slides of primary or recurrent breast cancer patients (added in Table S2). Since we attempted to show the correlation of HER2 with CD47 in Fig. 1 and no information on therapeutic history was obtained. However, we managed to obtain three paired tumors indicating the co-expression of HER2 and CD47 (Fig. 2e). Additional evidence show that CD47 was overexpressed in radioresistant breast cancer cell lines MCF7/C6 and 231/C5 compared to their primary cell lines; and flow cytometry analysis confirmed the enhanced population of CD47-expressing cells. Survival data from breast cancer database was also supportive to CD47 expression linked to the prognosis. Data in the revised manuscript Figs. 4b-4l showed that radioresistance could be reversed when CD47 was knocked out or blocked by antibodies. Together these results support our conclusion that enhanced CD47 expression is associated tumor aggressive growth and prognosis.

4. In Figure 3a the expression of HER2 by MCF7 is low (in some MCF7 cell lines is actually nil), while CD47 expression looks high and even higher than in the HER2-amplified BT474. In addition,

in Fig 3b the highest CD47 expression is observed in tumor T1, which has minor HER2 expression, and is absent in T2 that has the highest level of HER2 expression. The Authors should clarify whether the “HER2+ tumors” are actually amplified ones, and whether the samples were obtained from primary or metastatic lesions. The latter is particularly relevant in light of the data presented by the Authors in Fig 1 about the different expression of CD47 in primary and metastatic lesions.

Response: We very appreciate your insightful comments. It is true that the degrees of basal levels of HER2 and CD47 were hardly to be proportionally related based on the data of cell lines and individual tumors. However, we revealed that HER2⁻ BC cells or tumors showed CD47 deficiency or expressed CD47 at extremely low levels, whereas most of the HER2⁺ cells or tumors were enhanced CD47 levels (Figs. 1d-1f). To further confirm these findings, we have conducted more western blots of clinical samples of newly diagnosed breast cancer (8 HER2 positive and 8 HER2 negative). The new results showed the inconsistent correlation of HER2 and CD47 levels. Considering the results that HER2 induced CD47 expression via NF-κB signaling (Fig. 3), we suggest unknown mechanisms may exist in HER2⁺ cells which affect HER2-regulated CD47 protein levels and need to be studied in the future.

5. It is unclear to this Reviewer what the sentence on page 6 means (“Eight out of 13 HER2 positive tumors were CD47 high, 2 were CD47 low, and 3 showed a medium level; whereas, 10 of 13 HER2 negative tumors were low in CD47 expression, 3 showed medium, and none showed high CD47 expression (3c, right panel).”). Where these tumors are coming from? Are they part of the batch of 36 that are cited in 1? Again, what was their origin (primary lesion v. metastatic). Also, there is no logical link between the data illustrated in 3a-c, those presented in 1 about the role of irradiation and the “possibility that CD47 is enhanced in recurrent and metastatic breast cancer” as stated in lines 10-11 of page 6. CD47 is induced by IR and is found in HER2 overexpressing tumors (are they amplified?).

Response: Thanks again for pointing out. Since substantial data have been added and the logical link has been reorganized in the current version, we moved the previous Figs. 3c to Fig. 1f, and the corresponding descriptions in the Results and legends have also been rewritten in the current version. In Fig. 1f, the total number of breast cancer samples was 36, which consist of 18 HER2⁺ tumors and 18 HER2⁻ tumors, respectively. In Fig. 2e, another 36 samples, including 18 primary and 18 recurrent tumors were used for IHC staining which included three paired tumor samples. Samples in Fig. 1f were obtained from primary or recurrent breast cancer patients and the pathological slides in Fig. 2e included both HER2 positive and negative status (added in Table S2 and S3). CD47 was overexpressed in radioresistant breast cancer cell lines MCF7/C6 and 231/C5 compared to their primary cell lines, respectively. Flow cytometry confirmed this finding. Survival data from online databases also indicate that CD47 may confer the resistance. Data in the revised manuscript Figs. 4b-4l showed that radioresistance could be reversed when cd47 was knocked out or blocked by antibodies..

6. The data shown on Fig 3d-e are not particularly convincing, and it is unclear why the Authors selected the 231/C5 cells instead of using again the MCF7/C6 as in Fig. 1. Finally, what is the link between the experiment with irradiated 4T1 tumors (Fig. 3f) and the co-expression of CD47 and HER2 illustrated in the recurrent and primary tumors of Fig. 3g?

Response: Thank you for your comments. The MCF7/C6 cells-associated data have been added in the new Figs. 2b and 2d. In addition, induction of CD47 in diverse tumor cells in vitro and in vivo were also included in the revised manuscript with enriched CD47 proteins in HER2-expressing cells

and tumors (Figs. 1b-1f). Since CD47 was observed radiation-inducible in breast tumors, we hypothesized that CD47 may be expressed in high levels in radioresistant cells. Thus, radioresistant MDA-MB-231/C5 and MCF7/C6 cells, as well as the corresponding parental cells, were employed to determine the CD47 expression difference, HER2 was also detected in these cells for further confirming the relationship of HER2 status and CD47 expression (Fig. 2a,b). Both HER2 and CD47 could be significantly induced by radiotherapy in 4T1 tumors, which were inoculated in Balb/c mice with an intact immune system (Fig. 2c). These data strongly support that breast tumor cells may survive radiotherapy via enhanced HER2-mediated intrinsic pro-survival signaling and increased CD47-mediated immune evasion.

7. Given the relevance of the role of HER2 in the entire work, it is strongly suggested that IHC were adopting the criteria for HER2 scoring described in ASCO-CAP guidelines in 2013.

Response: We very appreciate this suggestion. The clinical data have been further reviewed and graded according to ASCO-CAP^[33] conducted by Dr. Yanhong Zhang, a pathologist at UC Davis Cancer Center. She is credited as a co-author for the revised manuscript. The scoring and grouping of clinical samples have been added in the Materials and Methods.

8. The sentence in the last line of page 6 starting with “Notably, both basal and IR-induced...” is repeated in the immediately following sentence. Please condense and rectify.

Response: Thanks. This sentence has been rewritten.

9. This reviewer can hardly see an effect of Lapatinib or Herceptin on C6 in Figure 4b. In 4c there is no effect of Herceptin on CD47 expression in BCSC exposed or un-exposed to irradiation. Also, there is a discrepancy between the CD47 basal expression in SKBR3 in Fig. 3a and that in Fig 4d, where it is elevated only after irradiation. This is not irrelevant given that the experiment is a key to show the link between HER2 amplification and CD47. The experiment in Fig. 4h and Fig. 4g again goes back to the radiation resistant MCF7/C6, but co-activation of HER2 and CD47 in IR-resistant cells was presented only for 231/C5 in Fig. 3d and Fig. 3e. Why this lack of consistency in the selection of the model cell line?

Response: We appreciate your pointing out. Indeed, although the basal levels among cell lines and patient tumors are varied, expression of CD47 and HER2 is obviously linked and inducible by radiation. We have repeated the experiments shown in Figs. 3a and 3b in the revised manuscript indicating that CD47⁺ population could be reduced by Lapatinib treatment. As shown in Fig. 3c, the basal CD47 expression was not affected by Herceptin treatment in HER2-expressing BCSCs, whereas the radiation-enhanced CD47 expression was impeded in the presence of Herceptin. Strikingly, the basal CD47 was extremely low in HER2-negative BCSCs, and could not be induced by radiation stimulation. To address the inconsistency in CD47 expression in the SKBR3 cells, we have repeated the experiments and the new data have been added. To further address the relationship of HER2 and CD47 regulation, we have established CRISPR/Cas9-human HER2 knockout cells and CRISPR/Cas9-human CD47 knockout cells, both from the MCF7/C6 cells. An array of new results obtained from these genetically edited radioresistant cells have added significant weights to our conclusion that these two receptors are mutually dependent in gene expression and that a synergy of anti both targets in immunotherapy is suggested (the data of CRISPR-Cas9 KO cells are added in the revised manuscript).

10. On page 7 the Authors state “The present data further revealed that the NF-κB-HER2-CD47

pathway can be induced by IR via NF- κ B activation, indicating the complexity of acquired tumor resistance.” However, this is not what the data in Fig. 4 show. There is no evidence as to the role of NF- κ B in the experiments illustrated in Fig. 4 and discussed in page 6 and 7 of the text. Overall the scheme sketched in Fig. 4i is tempting, but not demonstrated. Also, is it to be applied to bona fide HER2+ tumors that are almost always c-erbB2 amplified (SKBR3), or to IR resistant cell lines that bear co-activation of HER2 and CD47? Or to any cancer cell exposed to irradiation? The models presented in the different experiments correspond to completely different clinical conditions in man and open completely different therapeutic scenarios.

Response: We have conducted an array of experiments shown in Figs. 3f-k indicating that the promoter activation of both genes can be simultaneously regulated by NF- κ B that is well-defined in genotoxic stress including radiation. By application of the NF- κ B inhibitor IMD and mutations on the NF- κ B binding motifs in the CD47 promoter region we showed a reduced CD47 transcriptional activity via NF- κ B regulation. Chip assay further confirmed this in regulation.

We have revised the scheme sketched in Fig. 4i that is now shown in Fig. 7 with new supporting data. We have previously reported that radiation-induced NF- κ B activation is enhanced in HER2-expressing breast cancer cells^[7], and identifying HER2 promoter activity is controlled by NF- κ B regulation^[17]. We agree that the basal HER2 levels varied in a large scale. However, HER2 and CD47 are both inducible by radiation especially in mouse orthotopic tumors (Fig. 2c). It seems to be that no matter how the basal expression of HER2 (with gene amplification or not) or CD47, radiation activating NF- κ B will enhance both gene expression via promoter activation rather than gene copy amplification. Thus, as we have suggested before^[34], HER2 as well as CD47 status could be dynamically changeable in the cancer progression, and the status should be rechecked in recurrent and/or metastatic lesions, which may help to design more precise modalities to treat recurrent/metastatic cancers.

11. To this Reviewer 5e-g show that irradiation of MCF7 has no effect on phagocytosis, and that anti-CD47 actually lowers the phagocytosis itself. It is unfortunate that there are no data on MCF7 co-treated with antiCD47 and Herceptin. A similar pattern on phagocytosis is observed with MCF7/C6 and anti-CD47 (5f), or anti-CD47 and Herceptin (5g). How can this be used to conclude that irradiation “enhanced CD47 expression can compromise macrophage phagocytosis” (lines 7-8 of page 8) is frankly unclear to the reviewer. Do the Authors want to suggest that irradiation increased CD47 to such a level that that the antibodies are unable to block the down modulation of phagocytosis? This is not what could be expected from the evidence presented in the previous sections. Up to this point and to the evidence of Fig. 5 the hypothesis was that CD47 - the “don’t eat me signal”- was up-regulated with HER2 in irradiated cases, leading to immune evasion that would consist of avoidance of phagocytosis. Maybe the Authors should check that the conduct of the experiments caused an irradiation-dependent toxicity on macrophages. That would explain the findings and would call for a different experimental approach.

Response: We totally agree with the reviewer’s inquiry on why phagocytosis on irradiated cells were actually less than non-irradiated cells and why MCF7 cells that express less HER2 and CD47 than MCF7/C6 cells showed enhanced phagocytosis. This is a critical point for elucidating the fate of immune cells in an irradiated tumor microenvironment which should be further investigated. We believe that the possibility that radiation may damage the phagocytic function of macrophages is low since these experiments were conducted with macrophages that were not treated by radiation. To further confirm these findings, we have now added the new data of co-treatment with antiCD47 and

Herceptin that support a synergy of phagocytosis (Figs. 4e, 4f). Additionally, a new set of experiments has been conducted using CRISPR/Cas9-ko HER2 and CD47 cells (Figs. 4h-k).

12. With a somewhat logic leap forward, the Authors show in Fig 6 that indeed CD47 expression promoted the aggressive phenotype of MCF7/C6 that could be inhibited by anti-CD47. Importantly, the combination of irradiation, anti-CD47 and Herceptin, that did nothing on phagocytosis, eliminated clonogenic resistant cells. No idea of the mechanism of action for such an effect. Have the Authors any hypothesis to share?

Response: New data added in Figs. 4, 5, 6 further support that the combination of radiation with blocking either receptor or dual receptor inhibition could enhance the phagocytosis and elimination of radioresistant breast cancer cells. Additionally, we have added data with double KO of HER2 and CD47 that also enhanced the synergetic tumor inhibition shown in Fig. 5.

13. Experiments illustrated in Fig. 6f-h require some clarification. How large were the tumors at time 0 for the group exposed to anti-CD47 on D1? At week 1 they already were significantly smaller than those in the control group and in the anti-CD47 D15 (Fig.6f). The interpretation of the results of 6h is based on a crude difference of 20% survival in a single experiment involving 10 animals per group. The 2 more animals surviving is a very small difference for the conclusion about the synergy of low dose radiation and anti-CD47 therapy. Finally, it is unfortunate that there is no animal data for the triple combination of HER2 directed therapy, anti-CD47 and irradiation.

Response: A series of in vivo tests have been conducted shown in Figs. 5 and 6 with different combinations and dual gene knockouts. We have thoroughly revised the manuscript with details of protocol description tumor growth and treatments.

14. The Discussion has a number of statements that are not always supported by the data. In addition there are entire paragraphs (last of page 11 and first of page 12) that recapitulate the Results with direct reference to the figures in an unneeded rehearsal of data that were already extensively described in the Results. Here is a brief list of comments for the Authors:

a. As many other cancer treatment, irradiation causes cell death (including immunogenic cell death) and triggers mechanisms of resistance. The Authors show that CD47 expression may be one such mechanism that masks tumor cells from the phagocytic activity of macrophages. The hypothesis is sound, but the limiting factor of their observation is that combination of irradiation and therapies directed to CD47 and HER2 (Fig. 5) is far from being so active (already discussed in my point 10 above). The first paragraph of page 10 should take into account this aspect. Also, it is unclear what evidence in the present article supports the concept stated in lines 16-17 that "...a dual inhibition of CD47 and HER2 may enhance the abscopal effect of radiotherapy". The dual inhibition already works very little locally in the presented experiments. Based on what do the Authors expect an increased abscopal effect at a distance?

Response: (a) Again, we very appreciate your careful evaluation and your insightful comments on this work. The manuscript has been thoroughly revised and a substantial amount of new data especially the synergy of tumor inhibition by radiation combined with single or dual receptor blockade have been added. We agree that the target-inhibition studies, although informative, could not be able to clearly confirm the exact mechanism of cross-talk of these two receptors that can equip tumor cells different ability to survive radiation and regrow. Following this line of reasoning, we have vigorously reached the options of technologies and have now successfully established the dual CRISPR HER2 knockout and human CD47 knockout cells. In addition, tumor syngeneic breast

tumors which can reflect the immunotolerance by CD47-mediated anti-phagocytosis has also been added in the revised manuscript. All of these new data are supportive to the conclusion that these two receptors are closely regulated under radiation in radioresistant breast cancer cells. Thus, the potential therapeutic synergy is indicated by combined immunotherapy with inhibition of both targets, especially in the combined modality of radiation with immunotherapy in breast cancer treatments.

b. The sentence in page 10 line 21 “Recently cetuximab...” appears incomplete and out of context.

Response: This sentence has been corrected.

c. The parallel between the co- regulation of HIF1a and CD47 on page 11 has nothing to do with present work

Response: The HIF1a has been deleted.

d. The last sentence in the Discussion (“...local tumor radiotherapy with immune blockade of CD47 and HER2 may be an effective approach to enhance the tumor response to radiotherapy”) suggests that every irradiation to any type of cancer irrespective of the basal HER2 expression should be combined with anti-CD47 and anti-HER2 therapy. Frankly, this is not supported by the data.

Response: To further clarify the co-relationship of HER2 and CD47, in the revised manuscript, we have successfully generated a cell line from MCF7/C6 cells with HER2 knockout using CRISPR/Cas9 technology. The HER2-knockout cells show totally absence of HER2 expression indicating no off target effect and no HER2 expression by radiation. Supporting to our conclusion of radiation-induced CD47 dependent on HER2 expression level, no CD47 expression is induced by radiation in the HER2 CRISPR/Cas9 cells. Discussion of these new data has been added in the revised submission.

Literature cited in the response

1. Ivanov, VN, Bhoumik, A, Krasilnikov, M, Raz, R, Owen-Schaub, LB, Levy, D, Horvath, CM and Ronai, Z (2001) Cooperation between STAT3 and c-jun suppresses Fas transcription. *Molecular cell*, **7**: 517-28
2. Kortylewski, M, Xin, H, Kujawski, M, Lee, H, Liu, Y, Harris, T, Drake, C, Pardoll, D and Yu, H (2009) Regulation of the IL-23 and IL-12 balance by Stat3 signaling in the tumor microenvironment. *Cancer cell*, **15**: 114-23
3. Xu, Q, Briggs, J, Park, S, Niu, G, Kortylewski, M, Zhang, S, Gritsko, T, Turkson, J, Kay, H, Semenza, GL, Cheng, JQ, Jove, R and Yu, H (2005) Targeting Stat3 blocks both HIF-1 and VEGF expression induced by multiple oncogenic growth signaling pathways. *Oncogene*, **24**: 5552-60
4. Abell, K, Bilancio, A, Clarkson, RW, Tiffen, PG, Altaparmakov, AI, Burdon, TG, Asano, T, Vanhaesebroeck, B and Watson, CJ (2005) Stat3-induced apoptosis requires a molecular switch in PI(3)K subunit composition. *Nature cell biology*, **7**: 392-8
5. Hamilton, KE, Simmons, JG, Ding, S, Van Landeghem, L and Lund, PK (2011) Cytokine induction of tumor necrosis factor receptor 2 is mediated by STAT3 in colon cancer cells. *Molecular cancer research : MCR*, **9**: 1718-31
6. Dechow, TN, Pedranzini, L, Leitch, A, Leslie, K, Gerald, WL, Linkov, I and Bromberg, JF (2004) Requirement of matrix metalloproteinase-9 for the transformation of human mammary epithelial cells by Stat3-C. *Proceedings of the National Academy of Sciences of the United States of America*, **101**: 10602-7

7. Guo, G, Wang, T, Gao, Q, Tamae, D, Wong, P, Chen, T, Chen, WC, Shively, JE, Wong, JY and Li, JJ (2004) Expression of ErbB2 enhances radiation-induced NF-kappaB activation. *Oncogene*, **23**: 535-45
8. Guo, G, Yan-Sanders, Y, Lyn-Cook, BD, Wang, T, Tamae, D, Ogi, J, Khaletskiy, A, Li, Z, Weydert, C, Longmate, JA, Huang, TT, Spitz, DR, Oberley, LW and Li, JJ (2003) Manganese superoxide dismutase-mediated gene expression in radiation-induced adaptive responses. *Mol Cell Biol*, **23**: 2362-78
9. Yang, Y, Xia, F, Hermance, N, Mabb, A, Simonson, S, Morrissey, S, Gandhi, P, Munson, M, Miyamoto, S and Kelliher, MA (2011) A Cytosolic ATM/NEMO/RIP1 Complex Recruits TAK1 To Mediate the NF-kB and p38 Mitogen-Activated Protein Kinase (MAPK)/MAPK-Activated Protein 2 Responses to DNA Damage. *Molecular and Cellular Biology*, **31**: 2774-2786
10. Duru, N, Candas, D, Jiang, G and Li, JJ (2014) Breast cancer adaptive resistance: HER2 and cancer stem cell repopulation in a heterogeneous tumor society. *Journal of cancer research and clinical oncology*, **140**: 1-14
11. Ahmed, KM, Cao, N and Li, JJ (2006) HER-2 and NF-kB as the targets for therapy-resistant breast cancer. *Anticancer research*, **26**: 4235-4243
12. Kojima, Y, Volkmer, JP, McKenna, K, Civelek, M, Lusic, AJ, Miller, CL, Direnzo, D, Nanda, V, Ye, J, Connolly, AJ, Schadt, EE, Quertermous, T, Betancur, P, Maegdefessel, L, Matic, LP, Hedin, U, Weissman, IL and Leeper, NJ (2016) CD47-blocking antibodies restore phagocytosis and prevent atherosclerosis. *Nature*, **536**: 86-90
13. Hayden, MS and Ghosh, S. *Regulation of NF-kB by TNF family cytokines*. in *Seminars in immunology*. 2014. Elsevier.
14. Hoffmann, A, Levchenko, A, Scott, ML and Baltimore, D (2002) The IkappaB-NF-kappaB signaling module: temporal control and selective gene activation. *Science (New York, N.Y.)*, **298**: 1241-5
15. Wu, Y and Zhou, BP (2010) TNF-alpha/NF-kappaB/Snail pathway in cancer cell migration and invasion. *British journal of cancer*, **102**: 639-44
16. Karin, M and Lin, A (2002) NF-kB at the crossroads of life and death. *Nature immunology*, **3**: 221
17. Cao, N, Li, S, Wang, Z, Ahmed, KM, Degnan, ME, Fan, M, Dynlacht, JR and Li, JJ (2009) NF-kappaB-mediated HER2 overexpression in radiation-adaptive resistance. *Radiation research*, **171**: 9-21
18. Duru, N, Fan, M, Candas, D, Mena, C, Liu, HC, Nantajit, D, Wen, Y, Xiao, K, Eldridge, A, Chromy, BA, Li, S, Spitz, DR, Lam, KS, Wicha, MS and Li, JJ (2012) HER2-associated radioresistance of breast cancer stem cells isolated from HER2-negative breast cancer cells. *Clinical cancer research : an official journal of the American Association for Cancer Research*, **18**: 6634-6647
19. Poels, LG, Peters, D, van Megen, Y, Vooijs, GP, Verheyen, RN, Willemsen, A, van Niekerk, CC, Jap, PH, Mungyer, G and Kenemans, P (1986) Monoclonal antibody against human ovarian tumor-associated antigens. *Journal of the National Cancer Institute*, **76**: 781-91
20. Liu, X, Pu, Y, Cron, K, Deng, L, Kline, J, Frazier, WA, Xu, H, Peng, H, Fu, YX and Xu, MM (2015) CD47 blockade triggers T cell-mediated destruction of immunogenic tumors. *Nature medicine*, **21**: 1209-15
21. Chao, MP, Jaiswal, S, Weissman-Tsukamoto, R, Alizadeh, AA, Gentles, AJ, Volkmer, J, Weiskopf, K, Willingham, SB, Raveh, T, Park, CY, Majeti, R and Weissman, IL (2010)

- Calreticulin is the dominant pro-phagocytic signal on multiple human cancers and is counterbalanced by CD47. *Sci Transl Med*, **2**: 63ra94
22. Chao, MP, Tang, C, Pachynski, RK, Chin, R, Majeti, R and Weissman, IL (2011) Extranodal dissemination of non-Hodgkin lymphoma requires CD47 and is inhibited by anti-CD47 antibody therapy. *Blood*, **118**: 4890-901
 23. Tseng, D, Volkmer, JP, Willingham, SB, Contreras-Trujillo, H, Fathman, JW, Fernhoff, NB, Seita, J, Inlay, MA, Weiskopf, K, Miyanishi, M and Weissman, IL (2013) Anti-CD47 antibody-mediated phagocytosis of cancer by macrophages primes an effective antitumor T-cell response. *Proceedings of the National Academy of Sciences of the United States of America*, **110**: 11103-8
 24. Kim, D, Wang, J, Willingham, SB, Martin, R, Wernig, G and Weissman, IL (2012) Anti-CD47 antibodies promote phagocytosis and inhibit the growth of human myeloma cells. *Leukemia*, **26**: 2538-45
 25. Formenti, F, Constantin-Teodosiu, D, Emmanuel, Y, Cheeseman, J, Dorrington, KL, Edwards, LM, Humphreys, SM, Lappin, TR, McMullin, MF, McNamara, CJ, Mills, W, Murphy, JA, O'Connor, DF, Percy, MJ, Ratcliffe, PJ, Smith, TG, Treacy, M, Frayn, KN, Greenhaff, PL, Karpe, F, Clarke, K and Robbins, PA (2010) Regulation of human metabolism by hypoxia-inducible factor. *Proceedings of the National Academy of Sciences of the United States of America*, **107**: 12722-7
 26. Golden, EB, Pellicciotta, I, Demaria, S, Barcellos-Hoff, MH and Formenti, SC (2012) The convergence of radiation and immunogenic cell death signaling pathways. *Front Oncol*, **2**: 88
 27. Durante, M and Formenti, SC (2018) Radiation-Induced Chromosomal Aberrations and Immunotherapy: Micronuclei, Cytosolic DNA, and Interferon-Production Pathway. *Front Oncol*, **8**: 192
 28. Baccelli, I, Schneeweiss, A, Riethdorf, S, Stenzinger, A, Schillert, A, Vogel, V, Klein, C, Saini, M, Bauerle, T, Wallwiener, M, Holland-Letz, T, Hofner, T, Sprick, M, Scharpf, M, Marme, F, Sinn, HP, Pantel, K, Weichert, W and Trumpp, A (2013) Identification of a population of blood circulating tumor cells from breast cancer patients that initiates metastasis in a xenograft assay. *Nature biotechnology*, **31**: 539-44
 29. Baccelli, I, Stenzinger, A, Vogel, V, Pfitzner, BM, Klein, C, Wallwiener, M, Scharpf, M, Saini, M, Holland-Letz, T, Sinn, HP, Schneeweiss, A, Denkert, C, Weichert, W and Trumpp, A (2014) Co-expression of MET and CD47 is a novel prognosticator for survival of luminal breast cancer patients. *Oncotarget*, **5**: 8147-60
 30. Brightwell, RM, Grzankowski, KS, Lele, S, Eng, K, Arshad, M, Chen, H and Odunsi, K (2016) The CD47 "don't eat me signal" is highly expressed in human ovarian cancer. *Gynecol Oncol*, **143**: 393-397
 31. Ruiz-Saenz, A, Dreyer, C, Campbell, MR, Steri, V, Gulizia, N and Moasser, MM (2018) HER2 Amplification in Tumors Activates PI3K/Akt Signaling Independent of HER3. *Cancer research*, **78**: 3645-3658
 32. Merkhofer, EC, Cogswell, P and Baldwin, AS (2010) Her2 activates NF-kappaB and induces invasion through the canonical pathway involving IKKalpha. *Oncogene*, **29**: 1238-48
 33. Wolff, AC, Hammond, ME, Hicks, DG, Dowsett, M, McShane, LM, Allison, KH, Allred, DC, Bartlett, JM, Bilous, M, Fitzgibbons, P, Hanna, W, Jenkins, RB, Mangu, PB, Paik, S, Perez, EA, Press, MF, Spears, PA, Vance, GH, Viale, G, Hayes, DF, American Society of Clinical, O and College of American, P (2013) Recommendations for human epidermal growth factor receptor 2 testing in breast cancer: American Society of Clinical

Oncology/College of American Pathologists clinical practice guideline update. *Journal of clinical oncology : official journal of the American Society of Clinical Oncology*, **31**: 3997-4013

34. Geng, SQ, Alexandrou, AT and Li, JJ (2014) Breast cancer stem cells: Multiple capacities in tumor metastasis. *Cancer Lett*, **349**: 1-7

Reviewers' Comments:

Reviewer #3:

Remarks to the Author:

Candad-Green et al. Crosslink of CD47 and Her2 causes immunotolerance in radioresistant breast cancer.

This is very interesting manuscript that suggests the concurrent alterations of CD47 and HER2 may modulate breast cancer's immune-response after radiation therapy. It offers a novel mechanism and potential dual-targets to enhance immune-response to radiation therapy, and it explains some of the clinical association of treatment outcome with specific tumor types. The revision largely addressed the concerns of the previous Reviewer-1.

Perhaps my most concern was the choice of the word "crosslink" in the title and throughout the manuscript. This word is often used to refer a co-valent bond formation between two molecules. For example, in DNA repair field, it is used to specify a type of DNA damage such as DNA-protein crosslink, base-base crosslink, etc. In the context of this work, because CD47 and Her2 proteins do not form co-valent bond, but simply co-upregulated upon irradiation. The word "crosslink" seemed to be mis-used to capture the true finding of the manuscript, which was actually the co-upregulation in response to radiation, and that co-blockage of both receptors can enhance immuno-response.

Minor:

- Line 68: "CD47 a myeloid-specific immune checkpoint originally identified..." does not sound right. Perhaps it should be "CD47, a myeloid-specific immune checkpoint factor (or receptor), originally identified..."

Line 123: "Cop-transcriptional ..." should be "Co-transcriptional..."

Reviewer #4:

Remarks to the Author:

The submission of D. Candas et al is greatly modified manuscript from a previous version of 2016. The Authors show experiments that indicate a complex interplay between irradiation, NF-κB and its regulation of HER2 to sustain growth, and CD47 to evade innate immune surveillance.

The following points should be addressed by the Authors:

1. The sentence from line 79 to line 89 of the Introduction is a summary statement that repeats what is already presented in the abstract and in the Discussion. It should not appear in the Introduction

2. On lines 95 to 97 and the corresponding Fig 1c the Authors define MCF7 cells as HER2 expressing. MCF7 cells are used as negative control for HER2 expression, as also clarified by X.Dai, J Cancer. 2017; 8(16): 3131-3141. Indeed, the blot in fig 1d shows a very tenuous HER2 band for MCF7. Please explain the definition as HER2 expressing for MCF7

3. In Figure 1f CD47 expression is reported as much higher in HER2+ than in HER2-negative tumors. How many patients were considered in each group? What were the characteristics of breast cancers in the two groups in terms of estrogen and progesterone receptor status? How many of the patients in the HER2- group had a triple negative breast cancer?

4. In Fig 2f the curves on RFS and DMFS should also be shown for the groups CD47high/HER2 low and HER2high/CD47low. How the cutoff for high and low was selected?

5. The legend of Figure 3 states that "radiation enhanced CD47+ cells were reduced by antibody blockage of HER2.....". However, the experiments reported in Figure 3a and 3b show the effects if lapatinib, a small tyrosine kinase inhibitor, not an antibody. Please, take into account and discuss that lapatinib also blocks dimerization with HER2 and phosphorylation of EGFR, not only of HER2.

6. In Fig 5 the Authors show experiments of tumor growth with HER2-/-, CD47-/- and both CD47-/- and HER2-/- with (panel a) and without irradiation (panel c). In Fig 6 all experiments illustrated

in panels a, b and c with anti-CD47 and anti-HER2 therapies include irradiation. It is a very important control to show what happens also in the absence of irradiation, especially for the combination of anti-CD47 and anti-HER2. Consider that in man the combined block of CD20 with rituximab and of CD47 leads to responses in patients with lymphomas resistant to rituximab [Advani, R., et al. (2018). "CD47 Blockade by Hu5F9-G4 and Rituximab in Non-Hodgkin's Lymphoma." *New England Journal of Medicine* 379(18): 1711-1721]. This is not surprising given that rituximab and herceptin are IgG1 antibodies that sustain ADCC and are therefore very sensitive to the cooperation of macrophages. The analogies with the system illustrated in the manuscript are many and relevant. In particular, the proposed mechanism may be a more general stress-response mechanism that is not triggered solely by irradiation.

7. While the experiments with IMD0354 are intriguing and involve a possible role of NF- κ B, a more direct measure would be by use of a direct block of I κ B to rule out off target effects of the IMD0354 drug

8. The conclusion (last paragraph) allude to the possibility that every breast cancer tumor exposed to irradiation should receive dual block of HER2 and CD47 to avoid resistance. This is not completely supported by the data.

9. This Reviewer believes that the data presented in support of the main hypothesis of the manuscript should be organized in a different way:

a. CD47 and HER2 expression are often linked

i. Evidence in HER2 overexpressing tumors

ii. Evidence from radioresistant metastatic tumors irrespective of HER2 status of the primary tumor

b. In BC cell lines (and tumors) that are not characterized by HER2 overexpression irradiation leads to increased dual expression of HER2 and CD47 in cells surviving irradiation

c. The concerted increase of HER2 and CD47 is under the control of NF- κ B

d. Dual targeting of HER2 and CD47 synergizes with irradiation in HER2 non overexpressing tumors and in HER3 overexpressing and amplified breast cancer

Point-to-Point Response

Reviewers' comments:

Reviewer #3 (Remarks to the Author):

Candad-Green et al. Crosslink of CD47 and Her2 causes immunotolerance in radioresistant breast cancer.

This is very interesting manuscript that suggests the concurrent alterations of CD47 and HER2 may modulate breast cancer's immune-response after radiation therapy. It offers a novel mechanism and potential dual-targets to enhance immune-response to radiation therapy, and it explains some of the clinical association of treatment outcome with specific tumor types. The revision largely addressed the concerns of the previous Reviewer-1.

Perhaps my most concern was the choice of the word "crosslink" in the title and throughout the manuscript. This word is often used to refer a co-valent bond formation between two molecules. For example, in DNA repair field, it is used to specify a type of DNA damage such as DNA-protein crosslink, base-base crosslink, etc. In the context of this work, because CD47 and Her2 proteins do not form co-valent bond, but simply co-upregulated upon irradiation. The word "crosslink" seemed to be mis-used to capture the true finding of the manuscript, which was actually the co-upregulation in response to radiation, and that co-blockage of both receptors can enhance immuno-response.

Response: Thanks again for all your insightful comments and suggestions. The word "crosslink" in the title is indeed not properly representing the findings of this work. With additional new data added in the revised article, the title is modified as "Dual blockade of CD47 and HER2 eliminates radioresistant breast cancer cells". The word has been modified in the revised manuscript.

Minor:

- Line 68: "CD47 a myeloid-specific immune checkpoint originally identified..." does not sound right. Perhaps it should be "CD47, a myeloid-specific immune checkpoint factor (or receptor), originally identified..."

Response: Thanks for you point out. We have corrected the text in the revised manuscript. The corresponding texts are updated as: "CD47, a myeloid-specific immune checkpoint protein, originally identified as a component of the Rh blood group antigen complex is expressed in many cancer cells"

Line 123: "Cop-transcriptional ..." should be "Co-transcriptional..."

Response: Thanks, this typo has been corrected.

Reviewer #2 (Remarks to the Author):

The submission of D. Candas et al is greatly modified manuscript from a previous version of 2016. The Authors show experiments that indicate a complex interplay between irradiation, NF- κ B and its regulation of HER2 to sustain growth, and CD47 to evade innate immune surveillance.

The following points should be addressed by the Authors:

1. The sentence from line 79 to line 89 of the Introduction is a summary statement that repeats what is already presented in the abstract and in the Discussion. It should not appear in the Introduction.

Response: Thanks for your critical review and kind suggestions. We have rewritten the paragraph in Introduction with updated information on CD47 targeted immunotherapy. Also in the revised Introduction, we have added a brief summary of the key findings following the CD47 description as you suggested.

2. On lines 95 to 97 and the corresponding Fig 1c the Authors define MCF7 cells as HER2 expressing. MCF7 cells are used as negative control for HER2 expression, as also clarified by X. Dai, *J Cancer*. 2017; 8 (16): 3131–3141. Indeed, the blot in fig 1d shows a very tenuous HER2 band for MCF7. Please explain the definition as HER2 expressing for MCF7.

Response: Thanks again for pointing out this important point. It is true that MCF7 cells were used in many experiments as HER2 negative breast cancer cells in the work published by Dai et al. (*J Cancer*, 2017) ¹ and other publication regarding HER2 expression in MCF7 cells ²⁻⁴. With your suggestion, we further looked at the related publications cited in the work by Dai et al 2017, and found that no direct evidence was provided to define HER2 is negative but described as HER2 non-amplified, indicating that HER2 gene is impact whereas its inducible levels may be varied. In 2018, Slaga et al have published on *Sci. Trans. Med.* indicating that MCF7 is defined as a HER2 low-expressing breast cancer cell line that can be effectively eliminated by anti-HER2 therapy as effectively as HER2-overexpressing SKBR3 cells ⁵. In agreement, Turini et al reported that the binding activity of the Fab-like bispecific antibody targeting HER2 still remained efficient on low HER2-expressing MCF7 cells (HER2^{low}/IHC score 1+) ⁶. In addition, Novotny et al. described that MCF7 is a BC cell line expressing a modest level of HER2, and revealed that HER2 signaling in MCF7 cells was efficiently blocked by Lapatinib⁷. Also, Chung et al indicated that HER2 expression in MCF7 cells was elevated compared to triple negative breast cancer BT549 cells but lower than that in HER2-overexpressing SKBR3 and BT474 cells ⁸. In consistence with these observations reported in the literature, our results shown in Fig. 1b demonstrated that a certain basal level of HER2 expression is debatable in wild type MCF7 cells and can be significantly enhanced by radiation-induced NF κ B activation ⁹. These results indicate that the basal HER2 expression level in breast cancer cells including MCF7 is unstable and is highly sensitive to environmental alternations such as cell culture conditions and passage numbers, etc. Based on these observations, it seems reasonable to detect a certain low basal expression of HER2 in MCF7 cells.

3. In Figure 1f, CD47 expression is reported as much higher in HER2+ than in HER2- negative tumors. How many patients were considered in each group? What were the characteristics of breast cancers in the two groups in terms of estrogen and progesterone receptor status? How many of the patients in the HER2- group had a triple negative breast cancer?

Response: We very appreciate your insightful comments. To clarify the results, we have reorganized the figure 1f and presented as Figure 1d in the revised version. As shown in the legend of Figure 1d and referring the table attached here on the right, we described that “Left, representative images scored as low, moderate and high CD47 staining. Right, numbers of patients with low, medium or high IHC staining of CD47 grouped by HER2 positive or negative status (total HER2+ tumors n=18; total HER2- tumors n = 18)”. Specifically, according to the clinical information, the expression status of ER and PR of tumors used in this data were show in the right table. 10 (55.6%) of the 18 patients with HER2+ tumors had triple negative breast cancer.

4. In Fig 2f the curves on RFS and DMFS should also be shown for the groups CD47^{high}/HER2^{low} and HER2^{high}/CD47^{low}. How the cutoff for high and low was selected?

Response: To clarify the points, the RFS and DMFS for the groups CD47^{high}/HER2^{low} and HER2^{high}/CD47^{low} have been re-analyzed and the results are added in the revised Figure 2f which were the original Figures 1f and 1g. The data of OS in BC patients with lymph node metastasis or with endocrine therapy after surgery was thus moved to Figure S1a and S1b in the revised manuscript. To assess whether elevated HER2, CD47 or both could be related to OS, the median cutoff modus was applied following the published work on grouping patients with the median expression level. We have further revised the figure legend copied here as the following: “f Probability of recurrence-free (RFS) and (g) distant metastasis-free survival (DMFS) of BC patients of all subtypes stratified according to HER2 CD47 signature expression within HER2 strata from Breast Cancer Meta-base: 10 cohorts 22k genes database generated by SurvExpress (<http://bioinformatica.mty.itesm.mx:8080/Biomatec/SurvivaX.jsp>) from the HER2 probe 210930_s_at combined CD47 probe 211075_s_at. Statistical significance was analyzed by log-rank test”.

	HER2	ER	PR
Total	36 (100%)		
Negative	18 (50.0%)	16 (44.4%)	18 (50%)
Positive	18 (50.0%)	15 (41.7%)	13 (36.1%)
Unknown	0 (0.0%)	5 (13.9%)	5 (13.9%)

	Total	Number (%)
HER2+	HER2 ⁺ ER ⁺	10 (27.8%)
	HER2 ⁺ ER ⁻	6 (16.7%)
	HER2 ⁺ PR ⁺	11 (30.6%)
	HER2 ⁺ PR ⁻	5 (13.9%)
	HER2 ⁺ ER ⁺ PR ⁺	9 (25.0%)
	HER2 ⁺ ER ⁺ PR ⁻	1 (2.8%)
	HER2 ⁺ ER ⁻ PR ⁺	2 (5.6%)
	HER2 ⁺ ER ⁻ PR ⁻	4 (11.1%)
	HER2-	HER2 ⁻ ER ⁺
HER2 ⁻ ER ⁻		10 (27.8%)
HER2 ⁻ PR ⁺		2 (5.6%)
HER2 ⁻ PR ⁻		13 (36.1%)
HER2 ⁻ ER ⁺ PR ⁺		2 (5.6%)
HER2 ⁻ ER ⁺ PR ⁻		3 (8.3%)
HER2 ⁻ ER ⁻ PR ⁺		0 (0.0%)
HER2 ⁻ ER ⁻ PR ⁻		10 (27.8%)
ER/PR unknown		HER2 ⁺ ER ^{unknown} PR ^{unknown}

5. The legend of Figure 3 states that “radiation enhanced CD47⁺ cells were reduced by antibody blockage of HER2.....”. However, the experiments reported in Figure 3a and 3b show the effects of lapatinib, a small tyrosine kinase inhibitor, not an antibody. Please, take into account and discuss that lapatinib also blocks dimerization with HER2 and phosphorylation of EGFR, not only of HER2.

Response: We apologize for the inaccuracy. In the revised manuscript, we have thoroughly checked the terms of antibodies or small molecule inhibitors used for blocking CD47 or HER2. We have deleted the word of antibody in the legend of this figure, and with your kind suggestions, we have rewritten the result description with new citations on lapatinib and its function on HER2, which is copied here for convenience.

Lapatinib, a small-molecule kinase inhibitor that has also been reported to inhibit HER2, EGFR and HER3^{10,11} and approved for treatment of advanced metastatic BC patients. HER2-mediated activation of PI3K-AKT pathway causing NF-κB activation^{12,13}. We have previously reported that HER2-mediated AKT activation caused NF-κB leading to transactivation of HER2 promoter itself leading to the aggressive behavior of radioresistant BC cells⁹. During this work under review, recently, HER2 is shown to recruit AKT to disrupt STING pathway causing immunosuppression to virus infection¹⁴. Such HER2-mediated immunosuppressive function is demonstrated by HER2-induced CD47 upregulation in radioresistant BC cells. We found that blocking of HER2 by Lapatinib could efficiently reduce CD47⁺ population as shown in Figure 3a and 3b. The specific HER2-mediated CD47 expression was further identified by a totally absence of radiation-induced CD47 protein enhancement in HER2-expressing RD-BCSCs (Fig.3c).

6. In Fig 5 the Authors show experiments of tumor growth with HER2^{-/-}, CD47^{-/-} and both CD47^{-/-} and HER2^{-/-} with (panel a) and without irradiation (panel c). In Fig 6 all experiments illustrated in panels a, b and c with anti-CD47 and anti-HER2 therapies include irradiation. It is a very important control to show what happens also in the absence of irradiation, especially for the combination of anti-CD47 and anti-HER2. Consider that in man the combined block of CD20 with rituximab and of CD47 leads to responses in patients with lymphomas resistant to rituximab [Advani, R., et al. (2018). "CD47 Blockade by Hu5F9-G4 and Rituximab in Non-Hodgkin's Lymphoma." *New England Journal of Medicine* 379(18): 1711-1721]. This is not surprising given that rituximab and Herceptin are IgG1 antibodies that sustain ADCC and are therefore very sensitive to the cooperation of macrophages. The analogies with the system illustrated in the manuscript are many and relevant. In particular, the proposed mechanism may be a more general stress-response mechanism that is not triggered solely by irradiation.

Response: Thanks again for your insightful comments and the important information on you provided here. As this work has been conducted for a long term, accumulating new results using antiCD47 antibodies combined with other chemotherapeutic agents are being reported. Our original goal of this study was to define if radioresistant BC cells may enhance their immunosuppressive function which may compromise the immunotherapy efficacy. We agree with you that it is more informative to demonstrate the therapeutic effect of treatment with anti-CD47, anti-HER2 or the

combination of both in the absence of irradiation. Thus, although the in vivo experiments have been delayed almost for 4 month due to shelter in place, we have finished the experiments. The new mouse tests showed that indeed, tumor inhibition was more achieved by the dual receptor inhibition using in vivo antibody injection compared to single antibody treatment. However, the overall tumor growth curves were more inhibited in the groups combined with radiation especially in the group treated by RT with dual antibody inhibition of CD47 and HER2. These new data have been added to the revised manuscript (**Figure 6a-c**, **Supplementary Figure 10b-d**).

With your suggestions, we further searched the literature regarding CD47 blockade in synergizing with Rituximab to overcome aggressive and indolent lymphoma by enhancing

macrophage-mediated antibody-dependent cellular phagocytosis (ADCP)^{14, 15}. These are very important results on combination modalities of anti-CD47 and other anticancer antibodies. Based on their findings and other papers we read, as far no mechanistic work has been published on the crosstalk of CD47 and HER2 on the transcriptional regulation. However, there was no experimental evidence uncovered that the combination of anti-CD47 and anti-HER2 could significantly eliminates the radioresistant breast cancer cells by blocking their capability to escape macrophage surveillance. The mutual regulation of CD47 and HER2 due to NF- κ B signaling pathway in the cytoplasm may provide additional new insights on communications among cell surface receptors under genotoxic stress conditions. Thus, radioresistant cancer cells that are enriched with cancer stem cells could survive radiotherapy and may be aggressive due to enhanced ability of escaping immune surveillance^{16, 17}.

7. While the experiments with IMD0354 are intriguing and involve a possible role of NF- κ B, a more direct measure would be by use of a direct block of I κ B to rule out off target effects of the IMD0354 drug.

Response: Thanks for your insightful suggestion. IMD0354 is a selective IKK β inhibitor which blocks I κ B α phosphorylation in NF- κ B pathway. IMD-0354 was reported to specifically suppress the NF- κ B nuclear translocation¹⁸, with 98.5% inhibition on NF- κ B activity of NF- κ B at a concentration of 10 μ g/ml in HepG2 cells¹⁹. In our current study, we found that the activity of NF- κ B was significantly enhanced in wild type CD47-expressing cells with TNF- α stimulation while the TNF- α -induced enhancement could be blocked in cells expressing CD47 contains mutant NF- κ B binding motif in promoter region. We next employed IMD0354 for determining the NF- κ B-mediated CD47 expression regulation by luciferase reporter assay, fluorescence and western blot detection, and further confirmed the results by ChIP-PCR assay. Though it is convincing that we suggest NF- κ B regulates CD47 expression based on these experiments. It is reasonable to use more NF- κ B specific inhibitors to rule out off-target effects of IMD0354. Following your suggestion, we have additionally applied MLN120B (a specific ATP competitive IKK β inhibitor) and BMS-345541 (a selective inhibitor of the catalytic subunits of IKK) in the experiments for blocking NF- κ B activity. The new results showed that both MLN120B and BMS-345541 could efficiently inhibit the NF- κ B signaling-mediated CD47 transcription (**Supplementary Figure 4c, e and f**), which is in agreement with the finding that CRISPR-KO HER2 reduced the transcription of CD47 (**Figure 3f**). For further confirming the regulation of CD47 by HER2, Herceptin was used to treat cells transfected with CD47 wt or CD47 mut in the presence or absence of IR. Clearly, IR-induced CD47 expression could be inhibited by HER2 blockade (**Supplementary Figure 4d**). Regarding this point, the following are the summary of new data added to the revised manuscript which significantly strengthened the conclusion that CD47 transcription is regulated via HER2-NF- κ B pathway.

- 1) Added Supplementary Figure 4c, 4e, 4f, in which two additional selective IKK inhibitors were included for detecting the effect of NF- κ B signaling on CD47 transcription.
- 2) Added Supplementary Figure 4d, in which Herceptin was used to confirm the effect of HER2 on the transcription of CD47.

8. The conclusion (last paragraph) allude to the possibility that every breast cancer tumor exposed to irradiation should receive dual block of HER2 and CD47 to avoid resistance. This is not completely supported by the data.

Response: The conclusion has been revised to demonstrate a potential benefit of combined modality of RT with a dual blockade of CD47 and HER2 in eliminating the radioresistant BC cells especially in recurrent tumors expressing HER2.

9. This Reviewer believes that the data presented in support of the main hypothesis of the manuscript should be organized in a different way:

- a. CD47 and HER2 expression are often linked
 - i. Evidence in HER2 overexpressing tumors
 - ii. Evidence from radioresistant metastatic tumors irrespective of HER2 status of the primary tumor
- b. In BC cell lines (and tumors) that are not characterized by HER2 overexpression irradiation leads to increased dual expression of HER2 and CD47 in cells surviving irradiation
- c. The concerted increase of HER2 and CD47 is under the control of NF-kB
- d. Dual targeting of HER2 and CD47 synergizes with irradiation in HER2 non overexpressing tumors and in HER2 overexpressing and amplified breast cancer

Response: Again, we very much acknowledge your insightful review and the great effort for improving both of the scientific quality and data presentation for this work. Following your suggestions and through further discussion within the author team and based on the substantial new data generated, we have re-organized the figures in the revised version.

As you have noted, this study firstly introduced the expression features of CD47 and HER2 in diverse types of tumors. These results are looked after by studying the co-expression of CD47 and HER2 indicating the elevated expression levels in tumor tissues compared with the corresponding non-tumor tissues (TCGA data, **Fig. 1a** and **Supplementary Table 1**). Further study uncovered that the expression of CD47 was much higher in HER2⁺ cell lines than that in HER2⁻ cells. Consistently, elevated CD47 expression was observed in HER2⁺ tumors from BC patients compared with HER2⁻ BC tissues. Tumors with elevated CD47 levels were more frequently detected in the recurrent (with higher expression of HER2) versus primary tumors (with lower expression or negative status of HER2) in BC patients. Double high expression of HER2 and CD47 leads to worse prognosis of BC patients. On the basis of our published work which revealed that HER2 was irradiation-inducible in triple negative breast cancer cells²⁰, and considering both HER2 and CD47 were high expressed in recurrent tumor tissue (with higher expression of HER2), we detected the expression of these two proteins in radioresistant BC cells and found that both of HER2 and CD47 were high expressed in resistant cells compared with that in the corresponding parental cells. Since HER2 expression is sensitive to radiation and the expression of CD47 is irradiation-inducible in BC cells, an active crosstalk via NF-kB mediated signaling pathway could explain the mutually gene induction. Dual blockage of HER2 and CD47 is thus proposed to be an effective strategy synergizing with RT to eliminate the radioresistant BC cells under radiotherapy. The following are the summary of the re-organized data presentation in the revised manuscript.

- 1) The previous Fig. 1d has been moved to Fig. 1b
- 2) The previous Fig. 1e has been moved to Fig. 1c
- 3) The previous Fig. 1f has been moved to Fig. 1d
- 4) The previous Fig. 2e and supplementary Fig. 1c has been merged and moved to Fig. 1e.
- 5) The previous supplementary Fig1. a, b has been moved to supplementary Fig. 2b, c, respectively. The previous supplementary Fig. 1d has been deleted since the pictures were already shown in the revised Fig. 1e.

- 6) The previous Fig. 2f, g have been revised and moved to Fig. 1f, g, and the previous supplementary Fig. 3b, c have been moved to supplementary Fig. 1a, b, respectively.
- 7) The previous Fig. 1b, c have been moved to Fig. 2e, f, respectively.
- 8) The previous supplementary Fig. 2a, b have been moved to supplementary Fig. 1c, d, respectively.
- 9) The previous supplementary Fig. 3a has been moved to supplementary Fig. 2a.
- 10) The previous supplementary Fig. 4 has been renamed as supplementary Fig. 3
- 11) The previous supplementary Fig. 5a, b have been moved to supplementary Fig. 4a, b
- 12) The previous supplementary Fig. 6 has been renamed as supplementary Fig. 5
- 13) The previous supplementary Fig. 7 has been renamed as supplementary Fig. 6
- 14) The previous supplementary Fig. 8 has been renamed as supplementary Fig. 7
- 15) The previous supplementary Fig. 9 has been renamed as supplementary Fig. 8
- 16) The previous supplementary Fig. 10 has been renamed as supplementary Fig. 9
- 17) The previous supplementary Fig. 11 has been moved to supplementary Fig. 10a.

References cited for the response letter

1. Dai X, Cheng H, Bai Z, Li J. Breast Cancer Cell Line Classification and Its Relevance with Breast Tumor Subtyping. *J Cancer* 2017; **8**(16): 3131-41.
2. Lacroix M, Haibe-Kains B, Hennuy B, et al. Gene regulation by phorbol 12-myristate 13-acetate in MCF-7 and MDA-MB-231, two breast cancer cell lines exhibiting highly different phenotypes. *Oncol Rep* 2004; **12**(4): 701-7.
3. Zheng A, Kallio A, Harkonen P. Tamoxifen-induced rapid death of MCF-7 breast cancer cells is mediated via extracellularly signal-regulated kinase signaling and can be abrogated by estrogen. *Endocrinology* 2007; **148**(6): 2764-77.
4. Bacus SS, Kiguchi K, Chin D, King CR, Huberman E. Differentiation of cultured human breast cancer cells (AU-565 and MCF-7) associated with loss of cell surface HER-2/neu antigen. *Mol Carcinog* 1990; **3**(6): 350-62.
5. Slaga D, Ellerman D, Lombana TN, et al. Avidity-based binding to HER2 results in selective killing of HER2-overexpressing cells by anti-HER2/CD3. *Sci Transl Med* 2018; **10**(463).
6. Turini M, Chames P, Bruhns P, Baty D, Kerfelec B. A FcγRIII-engaging bispecific antibody expands the range of HER2-expressing breast tumors eligible to antibody therapy. *Oncotarget* 2014; **5**(14): 5304-19.
7. Novotny CJ, Pollari S, Park JH, Lemmon MA, Shen W, Shokat KM. Overcoming resistance to HER2 inhibitors through state-specific kinase binding. *Nat Chem Biol* 2016; **12**(11): 923-30.
8. Chung A, Choi M, Han BC, et al. Basal Protein Expression Is Associated With Worse Outcome and Trastuzumab Resistance in HER2+ Invasive Breast Cancer. *Clin Breast Cancer* 2015; **15**(6): 448-57 e2.
9. Cao N, Li S, Wang Z, et al. NF-κB-mediated HER2 overexpression in radiation-adaptive resistance. *Radiat Res* 2009; **171**(1): 9-21.
10. Gril B, Palmieri D, Bronder JL, et al. Effect of lapatinib on the outgrowth of metastatic breast cancer cells to the brain. *J Natl Cancer Inst* 2008; **100**(15): 1092-103.
11. Liu L, Greger J, Shi H, et al. Novel mechanism of lapatinib resistance in HER2-positive breast tumor cells: activation of AXL. *Cancer Res* 2009; **69**(17): 6871-8.
12. Ruiz-Saenz A, Dreyer C, Campbell MR, Steri V, Gulizia N, Moasser MM. HER2 Amplification in Tumors Activates PI3K/Akt Signaling Independent of HER3. *Cancer Res* 2018; **78**(13): 3645-58.
13. Merkhofer EC, Cogswell P, Baldwin AS. Her2 activates NF-κB and induces invasion through the canonical pathway involving IKKα. *Oncogene* 2010; **29**(8): 1238-48.
14. Wu S, Zhang Q, Zhang F, et al. HER2 recruits AKT1 to disrupt STING signalling and suppress antiviral defence and antitumour immunity. *Nat Cell Biol* 2019; **21**(8): 1027-40.

Reviewers' Comments:

Reviewer #2:

Remarks to the Author:

The manuscript follows a better organized description of the experiments and provides convincing evidence that co-expression of HER2 and CD47 is linked to mechanisms of breast cancer escape from phagocytosis and eventually eradication by irradiation. The additional experiments are clarifying some aspects of the NFkB modulation of the mechanism. The proposed mechanism is not fully proven but is consistent with the experiments and with what is known on HER2 and CD47. The Authors have amended their erroneous indication of lapatinib as a monoclonal antibody, except for referring to the small molecule as antibody in line 573 of the manuscript. Please change the sentence.

Point-to-Point Response

REVIEWERS' COMMENTS:

Reviewer #2 (Remarks to the Author):

The manuscript follows a better organized description of the experiments and provides convincing evidence that co-expression of HER2 and CD47 is linked to mechanisms of breast cancer escape from phagocytosis and eventually eradication by irradiation. The additional experiments are clarifying some aspects of the NFκB modulation of the mechanism. The proposed mechanism is not fully proven but is consistent with the experiments and with what is known on HER2 and CD47.

Response: We totally agree with your comments. The mechanistic interplay in NF-κB controlled crosstalk between CD47 and HER2 regulation is indeed to be further explored. Investigating the communication of these cell surface receptors may further reveal the integrated network governing cell growth and immune defending status.

The Authors have amended their erroneous indication of lapatinib as a monoclonal antibody, except for referring to the small molecule as antibody in line 573 of the manuscript. Please change the sentence.

Response: Thanks again for all your insightful comments and carefully reading which prevents the errors in our published work. The error on lapatinib description has been corrected.